

# Magnetic properties of pseudotachylytes, Jämtland, central Sweden

Hagen Bender[1], Bjarne S.G. Almqvist[2], Amanda Bergman[3], Uwe Ring[1]

[1]Department of Geological Sciences, Stockholm University , 106 91 Stockholm, Sweden
[2]Department of Earth Sciences, Geophysics, Villavägen 16, 752 36 Uppsala, Sweden
[3]Ramböll Sverige AB, Box 17009 , Krukmakargatan 21, 104 62 Stockholm, Sweden

*Correspondence to*: Hagen Bender (hagen.bender@geo.su.se)

## 1    Abstract

Nappe assembly in the Köli Nappe Complex, Jämtland, Sweden, has been associated with in- and out-of-sequence thrusting. Kinematic data from shear zones bounding the Köli Nappe Complex are compatible with this model, but direct evidence

from fault zones internally subdividing the nappe complex does not exist. We studied a series of pseudotachylyte exposures in these fault zones for deciphering the role seismic faulting played in the assembly of the Caledonian nappe pile. To constrain the fault kinematics, microstructural and magnetic fabrics of pseudotachylyte in foliation-parallel fault veins have been investigated. Because the pseudotachylyte veins are thin, we focused on small (c. 0.2 cm3) samples for measuring the anisotropy of magnetic susceptibility. The results show inverse proportionality between specimen size and anisotropy of

magnetic susceptibility degree, which is most likely an analytical artifact related to instrument sensitivity and small sample dimensions. This finding implies magnetic anisotropy results acquired from small specimens demand cautious interpretation. However, analysis of structural and magnetic fabric data indicates that seismic faulting occurred during exhumation into the upper crust but yield no kinematic in-formation. Structural field data suggest that seismic faulting was postdated by brittle E–W extensional deformation along steep normal faults. Therefore, it is likely that the pseudotachylytes formed late during

out-of-sequence thrusting of the Köli Nappe Complex over the Seve Nappe Complex.

## 2    Background

Pseudotachylytes are fault rocks that represent quenched frictional melts generated during co-seismic slip (Magloughlin and Spray, 1992; Sibson, 1975). Pseudotachylytes have been documented in fault zones within the Köli Nappe Complex, central Sweden (Beckholmen, 1982). Since the discovery of these localities, almost four decades ago, understanding of

pseudotachylyte generation has improved fundamentally (Lin, 2008; Rowe and Griffith, 2015). For example, pseudotachylyte characteristics have been used to infer dynamics of seismic faulting (e.g., Di Toro et al., 2005). A recently developed approach exploits magnetic properties and anisotropy of magnetic susceptibility of pseudotachylytes for deducing the focal mechanism of ancient earthquakes (Ferré et al., 2015). Here we adopt this method to better understand the kinematics of a ductile-to-brittle shear zone in the Köli Nappe Complex. Such information could offer direct evidence for



nappe stacking dynamics along this shear zone within the Köli Nappe Complex. These data are crucial for testing top-to-the-ESE, out-of-sequence fault propagation, which is indicated by map-scale nappe geometry (Beckholmen, 1984) and kinematic data collected elsewhere (Bender et al., 2018). Alternatively, the pseudotachylyte data can be compared to kinematic data from post-orogenic extensional faults crosscutting the nappe architecture (Bergman and Sjöström, 1997; Gee et al., 1994).

### 1.1   Rock magnetism and its application to pseudotachylytes

Frictional melting significantly affects magnetic properties of fault rocks. The magmatic mineral assemblage of pseudotachylytes commonly is distinctly different from that of its host rock (Ferré et al., 2012). Pseudotachylytes can contain magnetite produced during frictional melting (Nakamura et al., 2002). The rapid quenching leads to a remanent magnetization, which is acquired coseismically but sometimes contains post-seismic superimposed magnetizations, and hence impacts interpretation of paleomagnetism (Ferré et al., 2014; Fukuchi, 2003). Anisotropy of magnetic susceptibility of pseudotachylytes may also record information about the viscous flow of the friction melt (Ferré et al., 2015; Scott and Spray, 1999). Comparison of fault plane geometry and orientation with the magnetic fabric and petrofabric has been used to deduce earthquake kinematics and focal mechanism (Ferré et al., 2015).

The magnetic fabric of a rock is defined by its anisotropy of magnetic susceptibility (AMS), which in turn reflects the sum of individual magnetic responses of the rock-forming minerals (Borradaile, 1987). To use magnetic fabrics for inferring flow direction and sense, the carriers of rock magnetism must be known (Cañón-Tapia and Castro, 2004). In general, minerals respond in three fundamental ways to applied magnetic fields: diamagnetic, paramagnetic or ferromagnetic *sensu lato* (Butler, 1998; Tauxe, 2010). Depending on which of these behaviors is dominant in a rock specimen, AMS needs to be interpreted in different ways. Low susceptibility of diamagnetic minerals generally makes them subordinate contributors to the bulk rock AMS (Hirt and Almqvist, 2012, and references therein). Most rock-forming minerals are paramagnetic, for which AMS is foremost controlled by crystallography. AMS in paramagnetically-dominated rocks reflects the crystallographic preferred orientations of these minerals (Hirt and Almqvist, 2012). For ferromagnetic minerals, AMS is mainly controlled by grain shape and orientation distribution, with the exception of hematite (Borradaile and Jackson, 2010). With few exceptions, paramagnetic and ferromagnetic minerals express normal AMS fabrics. In such fabrics, the longest grain dimensions coincide with the maximum principal AMS axes (Tarling and Hrouda, 1993). For these cases the flow direction can be deduced from the magnetic lineation (Ernst and Baragar, 1992).

### 3   Regional geological context, field and macroscopic appearance of fault veins

In western Jämtland, the Köli Nappe Complex mainly consists of greenschist- to amphibolite-grade, calcareous metavolcanic and metasedimentary rocks exposed in the Tännforsen Synform (Figure 1; Beckholmen, 1984). Mineral and stretching lineations trend E–W to SE–NW. Foliations dip shallowly and their strikes follow the shape of the synform. Several minor



and two major fault zones separate the thrust sheets of the Köli Nappe Complex. The fault zones show ductile to brittle structures that are associated with pseudotachylyte (Beckholmen, 1982). We investigated the structurally highest of these fault zones, the Finntjärnen fault zone (Figure 1). The Köli Nappe Complex and its underlying units are crosscut by the Røragen Detachment and associated brittle, W-dipping normal faults (Bergman and Sjöström, 1997; Figure 1; Gee et al.,

1994). At Finntjärnen (63.389350N, 12.480276E), the schistosity of the micaschist host rock dips shallowly to the WNW (host rock schistosity $S_{hr}$ 302/15, Figure 2a). Mineral and stretching lineations carried by biotite and boudinaged amphibole crystals plunge shallowly to the W (host rock lineation $L_{hr}$ 262/08, Figure 2a). Mylonitic shear sense indicators associated with the ductile fabric were not observed. Fault veins commonly occur as foliation-parallel generation veins and crosscutting injection veins (Figure 2a, Figure 3).


The fault veins contain both pseudotachylyte and subordinate ultracataclasite. Macroscopically, four different types of fault rock can be distinguished: (1) **fractured fault rock** occurs in mm to cm wide domains between brittlely undeformed host rock and fault veins. It appears very similar to the host rock, features microscopic fractures and <1 mm to 10 mm-wide injection veins; (2) **cataclasite** has the same color as the host rock, but is much finer grained and appears as patches within

fault veins; (3) **pseudotachylyte** displays compositional flow-banding and consists of massive, bright gray, amorphous matrix containing <2 mm-sized clasts of host rock fragments and minerals; and (4) **altered pseudotachylyte** is bluish gray and massive, exhibits layer-parallel banding and generally sharp boundaries to unaltered pseudotachylyte (Figure 3). On slabs cut from an oriented sample (for details, see section 4.1), the coseismic slip direction cannot be deduced from fabrics in the fault vein. Deflection of compositional flow banding and asymmetric injection veins indicate top-to-the-E as well as top-

to-the-W sense of shear, respectively.
At the outcrop scale, offset schistosity planes and compositional layering in micaschist along brittle W-dipping, N–S striking faults, commonly occurs (Figure 2b). From orientations and slip sense of these faults, P- and T-axes indicating E–W extension were calculated (Figure 2b). These axes are a kinematic solution that give extensional (T) and contractional (P) directions of strain (Fossen, 2010). Calcite-filled veins crosscut the ductile fabric and the fault veins (Figure 3).

**4    Sampling, materials and analytical techniques**

**4.1    Sample preparation**

An oriented sample of a foliation-parallel fault vein with host rock on either side was collected in the field (sample AB15, schistosity $S_{AB15}$ 341/30, mineral lineation $L_{AB15}$ 269/07; Figure 2 and 3). The hand specimen was cut with a diamond saw into 5 to 10-mm-thick slabs parallel to the lineation and perpendicular to the foliation. A thin section was prepared from one

of these slabs. The rest of the slabs were cut into lineation-parallel sticks of approximately equal width and height. From these, 116 cube-shaped specimens were cut for magnetic experiments; x-axes of the cubes are parallel $L_{hr}$ and z-axes are





perpendicular to $S_{hr}$. Due to small spatial distribution of host/fault rock, specimen cubes are unconventionally small (side length $5.3 \pm 1.2$ mm, volume $0.17 \pm 0.13$ cm$^3$; uncertainty levels here and throughout the manuscript are 1σ) compared to standard-size specimens (7 to 11 cm$^3$) used in paleomagnetism (Table S1). Therefore, shape and size of cube dimensions

were compared to properties of the AMS ellipsoid and uncertainties related to the cube dimension were also investigated. Despite expending particular care in avoiding to cut specimens with different types of host/fault rock, specimens with mixed rock type occur. Approximate modal proportions for each rock type per specimen is presented in Table S2 in the electronic supplementary material.

### 4.2    Magnetic properties

### 4.2.1    Anisotropy of magnetic susceptibility

Anisotropy of magnetic susceptibility (AMS) was measured using a MFK1-FA susceptibility bridge (Agico, Inc.) operated at 200 A/m alternating current field and 976 Hz frequency. Orientation parameters used for data acquisition with Safyr4W software were P1 = P3 = 6 and P2 = P4 = 0 so that specimen x-axes plunge parallel to $L_{hr}$ and specimen z-axes point upward perpendicular to $S_{hr}$ ("Safyr4W User Manual," 2011). The AMS is expressed by the orientation and magnitude of the

principal axes of susceptibility $k_1 \geq k_2 \geq k_3$. Further parameters describing AMS data include the mean susceptibility $k_m = (k_1 + k_2 + k_3) / 3$, , magnetic foliation $F_m = k_2 / k_3$, magnetic lineation $L_m = k_1 / k_2$ and Jelinek's parameter for the degree of anisotropy $P_j$ (Jelinek, 1981). The shape of the susceptibility ellipsoid is described by $T = (\ln[F_m] - \ln[L_m]) / (\ln[F_m] + \ln[L_m])$. Only at $T = +1$ and $T = -1$ is the AMS ellipsoid oblate and prolate, respectively. For $0 < T < +1$, the AMS ellipsoid is more oblate than prolate; for $0 > T > -1$, it is in contrast more prolate (Jelinek, 1981). Mean

susceptibility $k_m$ has been normalized for specimen volume.

For data visualization, specimens containing more than one host/fault rock type were plotted based on their modal composition. The specimens were accounted to the dominant host/fault rock type composing the specimen. One specimen containing three rock types and 12 specimens composed of two rock types with each 50% mode were considered as mixed analyses. These data are therefore only presented in the data tables but were excluded from orientation and parameter

analysis of AMS data.

### 4.2.2    Frequency dependence of susceptibility

Frequency-dependent magnetic susceptibility was measured using a MFK1-FA susceptibility bridge (Agico, Inc.) operated at 200 A/m AC field and frequencies of $F_1 = 976$ Hz, $F_2 = 3904$ Hz, $F_3 = 15\,616$ Hz. In order to minimize the effect of anisotropy, all measurements were performed with the sample cubes oriented in the same position with their positive x-axes

horizontally pointing toward the operator (POS. 1 in "Safyr4W User Manual," 2011). Frequency dependence of susceptibility is characterized by the parameter $X_{FD} = 100(X_{LF} - X_{HF}) / X_{LF}$ (%) (Dearing et al., 1996).





Frequency dependence is used to identify superparamagnetic magnetite (grain sizes typically <20 nm), as this phase generally shows variation in bulk susceptibility as a function of field frequency. The method was here used to answer the question if very fine-grained magnetite formed during partial melting and recrystallization associated with the fault-slip that
formed the pseudotachylyte.

### 4.2.3   Temperature dependence of susceptibility

Temperature dependence of magnetic susceptibility was measured using a MFK1-FA, equipped with a CS4 furnace. Six sample cubes were analyzed individually; two sample cubes were analyzed together (AB15-13 and AB15-61) because of their small volumes. The samples were ground to a powder with an agate mortar, being careful not to contaminate the sample
with outside iron particles or magnetic phases from other materials. Magnetic susceptibility measurements at 200 A/m AC field and 976 Hz frequency were conducted from room temperature up to 700°C, and subsequently cooled back to room temperature, with a heating/cooling rate of 11.8 °C/min. Specimen AB15-67 was measured in air; all other specimens in argon atmosphere. Thermomagnetic data of the empty furnace were smoothed (5-point running mean) and subtracted from the sample thermomagnetic data using the Cureval8 software (Agico, Inc.).

### 4.2.4   Hysteresis

Hysteresis loops were performed with a LakeShore vibrating sample magnetometer with a maximum applied field of 1 T. Data processing was performed with the MATLAB toolbox HystLab (Paterson et al., 2018), using a linear high-field slope correction and automatic drift correction. The hysteresis data was normalized by the mass of the specimen. The extracted hysteresis parameters included saturation magnetization ($M_s$), saturation remanent magnetization ($M_{rs}$) and coercivity ($H_c$).

### 4.3   Shear sense determination using AMS

Obliquity between shear plane and magnetic fabric may be used to determine the sense of slip. Progressive shearing rotates maximum and intermediate principal axes of strain and AMS toward the shear plane (Borradaile and Henry, 1997). Kinematics are indicated in a plane perpendicular to the shear plane (i.e., fault vein margins) that contains the minimum and maximum AMS axes (cf. Figure 26 in Borradaile and Henry, 1997; and Figure 3 in Ferré et al., 2015). In this case, magnetic
foliations are inclined toward the slip direction, which gives the sense of shear.

## 5   Microstructural appearance of host and fault rocks

### 5.1   Host rock microstructure and petrography

Calcareous amphibole–biotite micaschist hosts the fault veins. Porphyroblastic biotite is oriented sub-parallel to the foliation (Figure 4a). Some grains show minor replacement by chlorite. Very fine-grained (<5 μm), euhedral Ti-oxides occur in the





center of patches where chlorite replaced biotite (Figure 5a). Amphibole is completely chloritized and only preserved as pseudomorphs (Figure 4a); their long axes have acute angles (<45°) to the foliation plane. The major opaque mineral is ilmenite (Figure 4a, Figure 5b). Ilmenite breakdown to Ti-oxide is observed at grain boundaries with biotite (Figure 5b). Boundaries between the brittlely undeformed and fractured host rock or fault or injection veins are sharp (Figure 4b). In the fractured host rock, alteration of biotite is more pronounced.

### 5.2    Fault rock microstructure and petrography

Cataclastic fault rock appears bright in thin section and consists of granular lithic and mineral fragments (Figure 4c). It forms bulky to drawn-out patches that grade into compositional flow banding in fault veins mainly composed of pseudotachylyte (Figure 3). Within cataclasite patches, tens to hundreds of µm thin, curved to meandering pseudotachylyte veins occur (Figure 4c). The modal abundance of cataclasite patches decreases from bottom to top of the studied fault vein (Figure 3).

Only fault rock with microstructural evidence for frictional melting is considered as pseudotachylyte. Such structural evidence includes microcrystallites, sulfide/oxide droplets and spaced survivor clasts, which may display embayed edges witnessing their partial melting (Magloughlin and Spray, 1992; Kirkpatrick and Rowe, 2013). All of these features are expressed in the studied pseudotachylytes. Sulfide/oxide droplets are submicron in size (Figure 4d). Grain size of survivor clasts is on the order of 20 µm to 100 µm (Figure 4d, Figure 5c–d). Their shapes are generally round although some exhibit

concave, serrated edges (Figure 5d). Quartz clasts are most common, although microcrystalline subhedral calcite partly appears radial around quartz clasts (Figure 4c). Furthermore, <5-µm-long needle-shaped crystals without obvious shape-preferred orientation occur dispersed in cryptocrystalline or amorphous matrix (Figure 5d). The needle-shaped microcrystallites are probably biotite, as energy-dispersive spectroscopic X-ray (EDS) mapping indicates they are enriched in Al, K, Fe and Mg as compared to the matrix. However, their small size prevented an interpretable single-crystal spectrum.

In some places, microcrystallites of unknown composition show dendritic patterns (Figure 5e).

A 4 to 10-mm-wide layer in the upper part of the studied fault vein exhibits a bubbly microstructure in transmitted light (Figure 4e–f). This spherically meandering microstructure represents chlorite alteration fronts replacing pristine pseudotachylyte (Figure 5e). The fine-grained (5 to 15 µm) chlorite displays no shape-preferred orientation. Where chlorite has replaced pseudotachylyte, micron- to submicron-sized, rhomb-shaped Ti-oxide crystals are finely dispersed (Figure 5f–

g). Their grain size decreases from center to rim of the chloritized domains.

Thin (<0.5 mm) antitaxial calcite + quartz veins with sharp edges cut across the host rock and all types of fault rock. They consist of mainly calcite and subordinate quartz. Euhedral pyrite occurs within such veins or in close proximity (<1 mm, Figure 3c). Vein orientations generally dip at high angles toward the W or are perpendicular to the foliation. Fibrous calcite, quartz and strain fringes around pyrite are compatible with E–W extension. The veins transect the boundaries between

different fault rock types and the host rock without being offset at these boundaries.





## 6    Rock magnetism results

### 6.1    Anisotropy of magnetic susceptibility

AMS data for all specimens are summarized in Table 1 and graphically presented in Figure 6. Magnetic anisotropy in host
rock and fault rock specimens displays consistent orientations of principal axes. Maximum principal axes ($k_1$) trend E–W and
are subparallel the host rock lineation for all rock types (Figure 6). Generally, all rock types show prolate AMS symmetry as
indicated by distribution of intermediate ($k_2$) and minimum principal axes ($k_3$) in a girdle perpendicular to $k_1$. Furthermore,
shapes and orientations of the 95% confidence regions for mean $k_2$ and $k_3$ axes reflect the prolate AMS shape (Figure 6a, c–
f). Symmetry of these confidence regions indicates homogeneous AMS fabrics for the analyzed specimen groups (Borradaile
and Jackson, 2010). However, intermediate and minimum principal axes for host rock specimens occur in two clusters
(Figure 6a). One cluster has $k_3$ axes perpendicular to the host rock foliation and $k_2$ axes lying within the foliation plane
(Figure 6b). The corresponding sub-fabric AMS ellipsoid approaches oblate shape ($T = 0.21 \pm 0.19$). The magnetic foliation
expressed by these specimens is subparallel to the schistosity $S_{hr}$. In the second cluster, $k_2$ and $k_3$ axes are inversely oriented.
Measurements of anisotropy ($P_j$, $T$) scatter over similar ranges for all rock types (Figure 7a). The anisotropy degree $P_j$ shows
highest variation for host rock specimens ($1.02 < P_j < 1.45$); lowest for altered pseudotachylyte ($1.06 < P_j < 1.25$). However,
the median $P_j$ values are similar in all rock types ($1.1 < P_j < 1.2$) and the middle 50% of these data overlap, when shown in
box-and-whisker plots (Figure 7b). The symmetry of the magnetic fabric shows no co-variation with the degree of anisotropy
(Figure 7a). Shapes of AMS ellipsoids for individual specimens of all rock types range from oblate to prolate (Figure 7c).
Overall, neither degree nor shape of the AMS ellipsoid define a magnetic fabric distinctive for one rock type or a group of
several rock types. Nevertheless, the volume-normalized mean susceptibility of altered pseudotachylyte specimens is
approximately twice as high (median $k_m = 4.7 \times 10^{-3}$ [SI]) as that of all other rock types (median $k_m = 2.7 \times 10^{-3}$ [SI]; Figure
8).

### 6.2    Temperature dependence of magnetic susceptibility

Thermomagnetic curves for heating and cooling of host rock, as well as for pristine and altered pseudotachylyte are
presented in Figure 9a–c. With increasing temperature, host rock thermomagnetic data exhibit steadily decreasing magnetic
susceptibility, followed first by a rapid increase to about twice the initial value at c. 500 °C, and then followed by a rapid
decrease at c. 580 °C (specimens AB15-115, AB15-116; Figure 9d). During cooling, host rock specimens show a prominent
rise in susceptibility at temperatures <600 °C and a peak at c. 430 °C. Pseudotachylyte specimens (AB15-12, AB15-13/61,
AB15-62) show a small but noticeable drop in susceptibility at 550–590 °C (Figure 9e). During cooling, susceptibility rises
sharply for all pseudotachylyte specimens at temperatures <590 °C (Figure 9b). Altered pseudotachylyte exhibits
progressively decreasing susceptibility with increasing temperature without any significant drop (specimens AB15-43,
AB15-67; Figure 9f). During cooling, susceptibility progressively increases to a peak at c. 300 °C and then gradually





decreases again. For specimens AB15-43, there is a small sharp increase of susceptibility at 590 °C observed in the cooling curve.

## 215    6.3    Hysteresis loops

Magnetic hysteresis measurements show all rock types respond dominantly paramagnetically to applied high magnetic fields (Table 2, Figure 10a, e, i). Hysteresis results for pseudotachylyte-free specimens show either no or very minor ferromagnetic response. They have saturation magnetizations ($M_s = 2.3 \pm 1.3 \times 10^{-4}$ Am$^2$ kg$^{-1}$) about one order of magnitude below those specimens containing pseudotachylyte ($M_s = 1.73 \pm 0.6 \times 10^{-3}$ Am$^2$ kg$^{-1}$) (Table 2). Furthermore, pseudotachylyte-free
specimens have generally very open slope-corrected hysteresis loops, which do not display branches characteristic of ferromagnetic minerals (Figure 10b, j) (cf. Paterson et al., 2018). Slope-corrected hysteresis curves for these specimens accordingly also display atypical shapes, which may result from an artificial correction to the data (Figure 10c, k). Contrastingly, hysteresis loops for pseudotachylyte-bearing specimens show a ferromagnetic contribution in magnetic response. This is expressed weakly in the unprocessed hysteresis loop (Figure 10e), and more clearly after linear high-field
slope correction (Figure 10f; although hysteresis loops generally fail to close at high fields). Based on hysteresis parameters, pristine pseudotachylyte-rich specimens have the highest saturation remanent magnetization ($M_{rs} = 3.9 \pm 1.8 \times 10^{-4}$ Am$^2$ kg$^{-1}$), compared to host rock ($M_{rs} = 6.6 \pm 7.5 \times 10^{-5}$ Am$^2$ kg$^{-1}$) and altered pseudotachylyte specimens ($M_{rs} = 1.2 \pm 0.5 \times 10^{-4}$ Am$^2$ kg$^{-1}$). Magnetic hysteresis raw data is available in Table S3 in the electronic supplementary material.

## 230    6.4    Specimen size and shape

Specimen cube dimensions deviate moderately from neutral shapes. Their long edges are between 4.1% and 20.9% longer than their short edges. Prolate and oblate shapes are equally common (Figure 11a, Table S1). The shape parameters of specimen dimensions ($T_d$) and magnetic anisotropy ($T$) are independent of each other (Figure 11b, Table S1). The degree of anisotropy of specimen shape and magnetic susceptibility show no significant correlation (Figure 11c).
Raw measurements of mean susceptibility ($k_m$) and anisotropy degree ($P_j$) are inversely proportional (Figure 12a). Additionally, the standard error of $k_m$ decreases with increasing specimen volume (Figure 12b). Consequently, the AMS data are dependent on specimen size. Small specimen volumes result in larger uncertainties, which in turn causes higher $P_j$ values. This observation is further discussed in section 7.5 which also discusses the limitation of specimen size in studies using AMS.





**7    Discussion**

**7.1    Source of magnetic susceptibility and its anisotropy**

Thermomagnetic heating curves for host rock specimens show decrease in magnetic susceptibility with increasing temperature until 400 °C, which is characteristic of paramagnetic behavior (Figure 9d) (Hunt et al., 1995). Formation of new magnetite at temperatures above 400 °C is indicated by the peak and sudden decrease in magnetic susceptibility at 580 °C,

the Curie temperature of magnetite (Hunt et al., 1995).  These results, together with magnetic hysteresis data (Table 2, Figure 10), show that the magnetic susceptibility of the host rock micaschist arises from paramagnetic minerals. It follows that the AMS in the host rock is controlled by the crystallographic orientation of the paramagnetic minerals (Borradaile and Jackson, 2010). An AMS sub-fabric in host rock specimens has parallel magnetic and mineral lineations and sub-parallel magnetic and ductile foliations (Figure 6b). Shape-preferred orientation of tabular biotite crystals in the host rock (Figure 4a) implies

crystallographic $c$-axes of biotite are oriented perpendicular to the schistosity. This AMS sub-fabric is therefore inferred to originate from crystallographic preferred orientation of biotite, which in single crystals exhibits $k_3$ axes subparallel to biotite crystallographic $c$-axes (Borradaile and Henry, 1997; Martín-Hernández and Hirt, 2003). The mean magnetic susceptibility $k_m$ of host rock specimens ($k_m = 2.62 \pm 0.46 \times 10^{-4}$, SI) is in the range of typical of schists (km = 0.026–3.0 × 10−3 [SI], Hunt et al., 1995). Single-crystal bulk susceptibility of biotite, muscovite and chlorite are on the same order of magnitude

around $10^{-4}$ [SI] (Martín-Hernández and Hirt, 2003). In the absence of magnetite, the host rock AMS most likely arises from these sheet silicates.

Fractured host rock and cataclasite specimens without pseudotachylyte display the same magnetic properties as host rock specimens (Figures 7, 8, Tables 1, 2). This conformity suggests the same paramagnetic source of the AMS with contributions from biotite, white mica and chlorite.

Pseudotachylyte thermomagnetic data shows a distinct drop in susceptibility from 550 °C to 590 °C, which indicates presence of magnetite (Figure 9e). Hysteresis results of pseudotachylyte-bearing specimens show mixed paramagnetic and ferromagnetic behavior (Table 2, Figure 10e–g). The AMS of pseudotachylytes thus reflects the sum of paramagnetic and ferromagnetic minerals in these specimens. The narrow range of $k_m$ does not offer the opportunity to isolate sub-sets (Table 1, Figure 8), which is a common approach to separate AMS sub-fabrics caused by paramagnetic and ferromagnetic minerals

(Borradaile and Jackson, 2010). The presence of magnetite does not seem to increase $k_m$ to values significantly higher than the (fractured) host rock and/or cataclasite specimens (Figure 8). The ferromagnetic contribution to the pseudotachylyte AMS is consequently small. The pseudotachylyte AMS is therefore likely controlled by crystallographic preferred orientation of its paramagnetic minerals, that is most probably biotite, with a subordinate contribution from the shape preferred orientation of magnetite (cf. section 5.2).

In altered pseudotachylyte specimens, successive decrease in magnetic susceptibility without significant drop at 580 °C during heating indicates dominant paramagnetic behavior. This behavior suggests that magnetite present in pristine pseudotachylyte has been altered to an unknown phase in chloritized pseudotachylyte (Figure 9f). Magnetic hysteresis results



confirm bulk paramagnetic behavior for altered pseudotachylyte (Table 2, Figure 10i–k). The mean magnetic susceptibility for altered pseudotachylyte being about twice as high as that for other rock types (Table 1, Figure 8b), the AMS of altered pseudotachylyte apparently has an additional or a different mineral source than the other rock types. Bulk magnetic susceptibility for single-crystal chlorite without high-susceptibility mineral inclusions is about twice that of biotite and muscovite single crystals (Martín-Hernández and Hirt, 2003). These sheet silicates were also argued to collectively cause AMS in host rock specimens, but in altered pseudotachylyte chlorite is much more abundant, making up to c. 50% of the mode (Figure 4e–f, Figure 5e–g). We infer that AMS in altered pseudotachylyte dominantly reflects the orientation distribution of chlorite.

### 7.2 Petrofabric versus magnetic fabric orientations

Fault vein margins give the orientation of the slip plane in the Finntjärnen fault zone. Seismic faulting in these veins occurred parallel to the schistosity $S_{hr}$ along subhorizontal, shallowly W-dipping shear planes (Figure 2, Figure 3). The slip direction is indicated by subhorizontal E–W trending magnetic lineation in all fault rock types (Figure 6). This direction is consistent with mineral and stretching lineations expressed in the ductilely deformed host rock. These orientations also coincide with the extension direction defined by crosscutting normal faults (Figure 2b). Obliquity between the pseudotachylyte magnetic foliation and fault vein margins would indicate the kinematics of seismic slip (Ferré et al., 2015). However, the AMS of both cataclastic and friction melt-origin fault rocks displays prolate symmetry and magnetic lineations that are parallel with the vein margins. These results show that neither a magnetic foliation nor obliquity with the shear plane are developed, as would be expected from noncoaxial deformation (Borradaile and Henry, 1997). The observed AMS data raises several questions: (1) If such a kinematic model does not agree with the observed fault rock AMS, what process aligned the maximum principal axes? (2) How is it possible to explain the distribution of intermediate and minimum principal axes in a girdle perpendicular to the $k_1$ axes? (3) Why are AMS fabrics of all rock types compatible? Based on qualitative observations, preferred orientation of pseudotachylyte microcrystallites as well as chlorite in altered pseudotachylyte is absent. Shape-preferred orientation, and thus crystallographic-preferred orientation of the tabular paramagnetic carriers of magnetic susceptibility offers no direct explanation.

### 7.3 Deformation sequence

Foliation-parallel fault veins, bound by narrow domains of fractured host rock, crosscut the ductile host rock fabric (Figures 2–4). Their formation thus postdated ductile upper-greenschist to amphibolite facies deformation, which is in line with previous work (Beckholmen, 1982; 1983; 1984). Whether chloritization of amphibole occurred before or after fault vein formation is not evident. The fault veins contain unmolten cataclasite, frictional melt-origin pseudotachylyte and altered pseudotachylyte in varying modal amounts. Spaced survivor clasts, microcrystallites and submicron sulfide/oxide droplets in pseudotachylyte identify these fault rocks as quenched, coseismic friction melts (Figure 4, Figure 5) (Magloughlin and Spray, 1992; Cowan, 1999; Rowe and Griffith, 2015). Chloritization of the pseudotachylyte groundmass, and pronounced



replacement of biotite to chlorite in fractured host rock domains indicate that hydrothermal alteration was associated with faulting. The chlorite microstructure suggests that recrystallization was static (Figure 5). After pseudotachylyte formation, ambient temperature conditions in the fault zone are therefore inferred to be of lower greenschist-facies (cf. Di Toro and Pennacchioni, 2004; Kirkpatrick et al., 2012). We deduce seismic faulting and subsequent alteration of fault rocks in the Finntjärnen fault zone occurred in the brittle–ductile transition zone near the base of the upper crust. Assuming the typical

temperature range of 300–350 °C, and depending on the thermal gradient, the faulting occurred at c. 12 ± 4 km depth (cf. Sibson and Toy, 2006).

Brittle faults and fibrous calcite + quartz veins crosscut both the ductile host rock fabric and the fault veins at high angles. Their orientations relative to the fault vein geometry, together with microscopic and macroscopic observations (Figures 2–5), suggest that these E–W extensional structures formed latest. These structures are consistent with other extensional structures

related to the Røragen Detachment west of the Tännforsen Synform (Figure 1) (Gee et al., 1994; Bergman and Sjöström, 1997). In summary, seismic faulting in the Finntjärnen fault zone occurred after the formation of the upper greenschist-/amphibolite-facies schistosity, and prior to late-stage E–W extensional brittle structures. Structural overprinting relations imply transport of thrust sheets in the Köli Nape Complex during exhumation of these nappes from the middle to the upper crust. The sense of faulting can, however, not be deduced from the here presented data. Nevertheless, previous work in the

area indicated that thrusting was toward the ESE (Bergman and Sjöström, 1997; Bender et al., 2018).

### 7.4    Regional tectonic implications

Structural and magnetic analysis of pseudotachylyte-bearing fault veins and their ductilely deformed host rocks reveals that petrofabric and AMS are co-parallel. The accordance of these data might indicate that ductile host rock fabrics and brittle fault rock fabrics developed in the same strain field. However, the orientations of AMS and petrofabric in host rock versus

fault rock specimens could not be used for deducing the kinematics of neither ductile nor seismic shear. Nevertheless, crosscutting relations show that pseudotachylites formation in the Finntjärnen fault zone predated E–W extensional deformation. Implications on nappe stacking kinematics in the Köli Nappe Complex can nevertheless not be drawn and are left for further investigations. An additional aspect worth clarifying are crosscutting relations in the footwall of the extensional Røragen Detachment in the west of the Tännforsen Synform (Figure 1).

### 330   7.5    Methodological remarks on AMS of small specimens

There is an apparent inverse relationship between $k_m$ and $P_j$, as well as a linear relationship between degree of anisotropy and standard error. This effect is caused by specimen size. The larger specimens (by volume) have in general higher bulk susceptibility, and $P_j$ tends towards lower values ranging from 1.01 up to 1.10. Normalization for specimen volume has little impact in removing this bias and it is therefore evident that specimens with very small size are more likely to produce a large

scatter in the degree of anisotropy. Although this is an undesired artifact, it demonstrates the limitation of using small sample



cubes in the current setup with the MFK1-FA system. The effect is furthermore emphasized by the increase in $k_m$ standard error as a function of $P_j$.

Observations of magnetic anisotropy made in this study raises the issue of measuring AMS of specimens with small volume. Current equipment that exists commercially are not designed for handling small specimen volumes and in most applications

the intended volume ranges from 7 to 11 cm$^3$ (representing standard size cubes and cylinders used in paleomagnetic and AMS studies). However, there is a growing interest for measurements of small specimens, as many AMS studies target geological structures that occur on the cm to sub-cm scale (e.g., Ferré et al., 2015). One of the challenges in using smaller specimens is clearly an increased uncertainty in manufacturing specimens that have appropriate dimensions. However, specimens can be constructed with care to compensate for this effect, and in this study, we have demonstrated that the non-

equidimensional effect is secondary in importance to the specimen volume. Furthermore, our AMS data show a consistent magnetic fabric in the different rock types, which suggests that they most likely represent the true rock fabric (although the magnitudes are variable).

## 8    Conclusions

Field, microstructural and magnetic fabric data from the Finntjärnen fault zone provide the following constraints on seismic

faulting recorded by pseudotachylyte-bearing fault veins:

(1) Structural overprinting relations show seismic faulting occurred during exhumation of the Köli Nappe Complex into the upper crust within the seismic zone, and before brittle E–W extension.

(2) Neither the petrofabric nor magnetic fabrics reveal the coseismic slip direction.

(3) Chloritization of pseudotachylyte resulted in higher bulk magnetic susceptibility as compared to pristine

pseudotachylyte. The very low amount of magnetite in pseudotachylyte, although detectable by its magnetic behavior based on thermomagnetic curves, hysteresis loops and bulk susceptibility, does not contribute substantially to the pseudotachylyte bulk magnetic behavior.

(4) Unconventionally small specimen size increases the degree of anisotropy of magnetic susceptibility measurement data. Magnetic anisotropy results in small specimens demand cautious interpretation, but offers a promising new

venue to study detailed geological features.

## 9    Data availability

All data that led to the conclusions of this paper are presented in the figures, tables and supplementary material.

## 10    Supplement

**Table S1: Mass, dimensions, volume and dimension parameters for individual specimen.**

Please see separate file 'TableS1SpecimenMassSize.xlsx'





**Table S2: Approximate proportions of different host and fault rock types for individual specimen.**

Please see separate file 'TableS2RockTypeProportions.xlsx'

**Table S3: Magnetic hysteresis raw data.**

Please see separate file 'TableS3HysteresisRawData.xlsx'

## 11  Author contribution

Field work was carried out by HB and AB. HB and BSGA conducted the magnetic experiments, processed and interpreted the results. HB created figures and tables and wrote the initial draft, which was edited by all co-authors.

## 12  Competing interests

The authors declare that they have no conflict of interest.

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



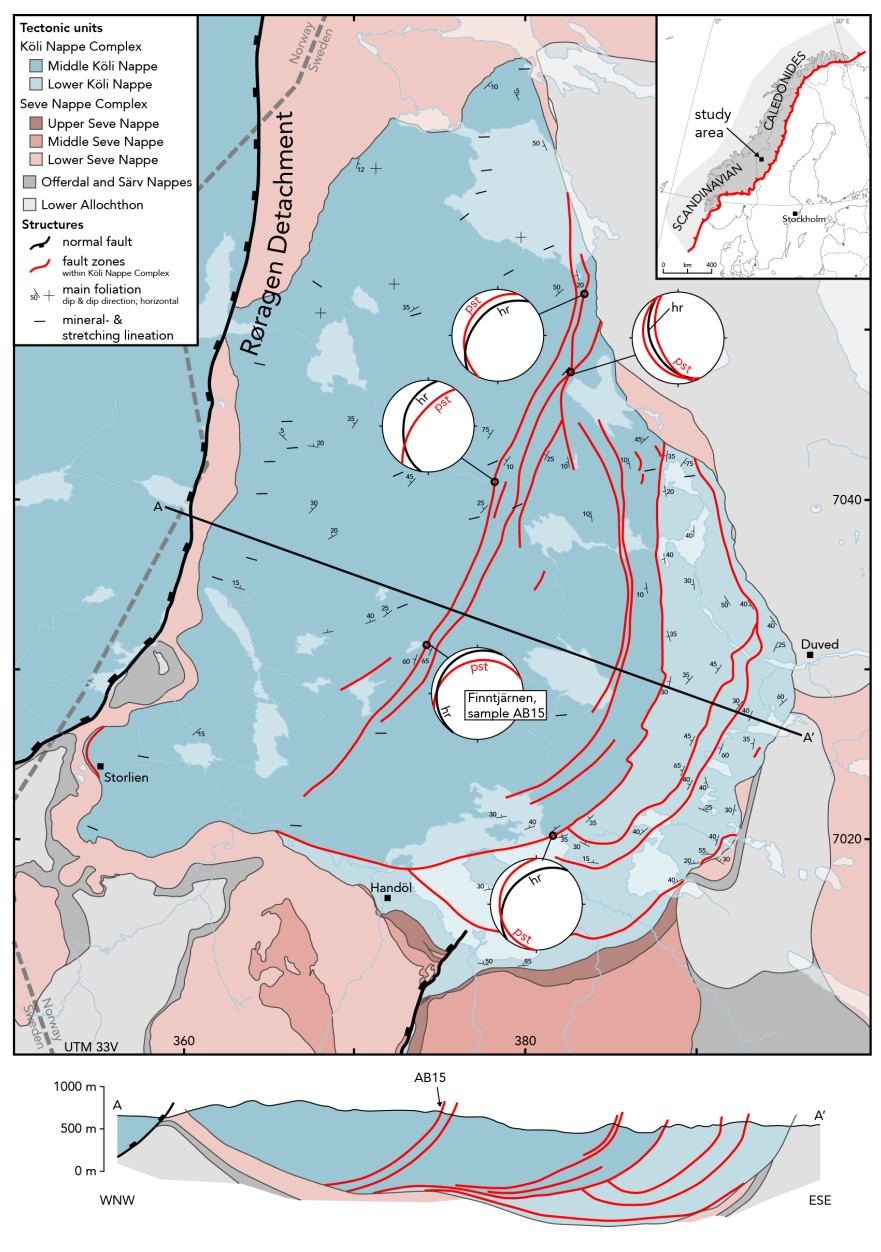






**Figure 1. Geological map of the Tännforsen Synform and section A–A′ across it (modified after Beckholmen, 1984). Structural position of tectonic units is indicated in the top left. Structurally lower faults are truncated by structurally higher faults within and beneath the Köli Nappe Complex. Stereographic projections (lower hemisphere, equal-area) show orientations of host rock schistosity (hr) sub-parallel to pseudotachylyte-bearing fault veins (pst) (data from this study and Bergman, 2017). The Røragen Detachment in the west cuts across all other units beneath it, illustrating that it developed structurally latest.**




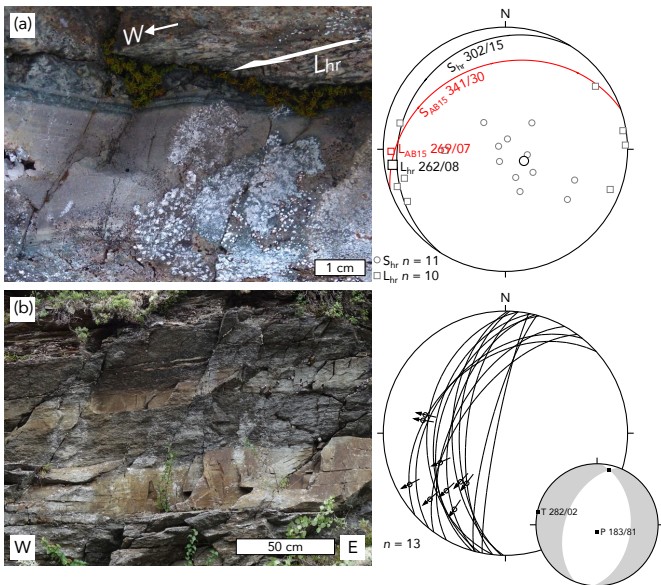

**Figure 2. (a) Field photograph of the microstructurally investigated fault vein. It is 3 cm wide, foliation-parallel and exhibits a 5-mm-thick band of bluish, altered pseudotachylyte at its top. Note the crosscutting, steeply dipping fractures. Equal-area projections with poles to host rock schistosity planes ($S_{hr}$), host rock lineations ($L_{hr}$) and orientation of the investigated sample AB15 (red great circle). Mean orientations for $S_{hr}$ and $L_{hr}$ are indicated with large symbols. (b) Brittle, steeply W-dipping normal faults crosscut the ductile fabric. Sense of slip is indicated by calcite slickenfibres on some of the fault planes. Fault plane solution for these faults (bottom left) shows E–W extension (data processed with FaultKin 7, Marrett and Allmendinger, 1990).**



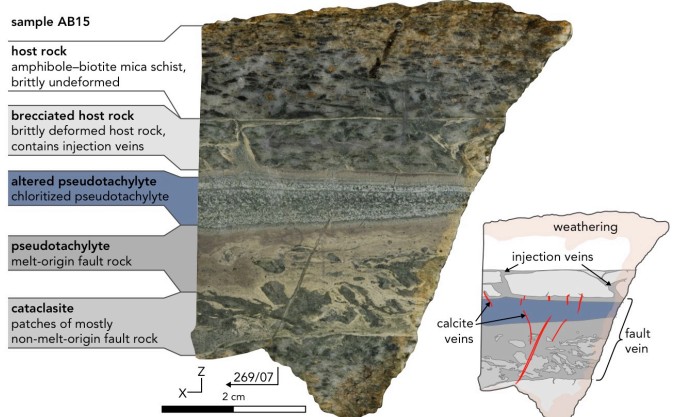

**Figure 3. Macroscopic appearance of a foliation-parallel fault vein exhibiting different kinds of fault rock. For detailed description, see text. Characterization of fault rock types is also based on microscopic observations. The image represents the XZ**
**plane of the ductile finite strain ellipsoid.**



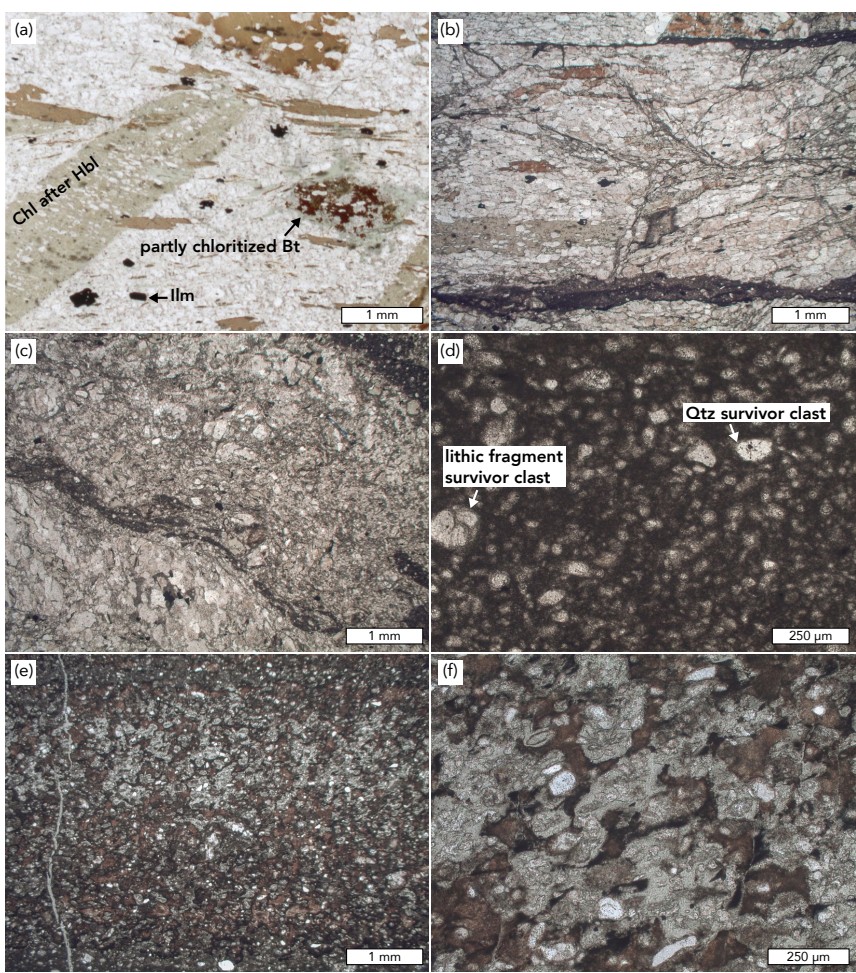

**Figure 4. Microstructural appearance of host rock and different fault rock types (plane polarized light). (a) Host rock: foliated micaschist with fine-grained (50–200 µm) quartz + white mica + plagioclase matrix and 1 to 10 mm big porphyroblasts of biotite (fresh: top, partly chloritized: middle right) and pseudomorphs of chlorite after amphibole (left). (b) Brittlely deformed domain in the host rock with multiple fractures filled with cataclasite (center) and/or pseudotachylyte (top and bottom). (c) Cataclastic fault rock: host rock fabric is completely obscured, some lithic fragments remain (bottom left), pseudotachylyte veins occur with patchy and diffuse borders (center, top right corner). (d) Pseudotachylyte: cryptocrystalline matrix containing 20–100-µm-sized, spaced survivor clasts of quartz and lithic fragments. (e–f) Spherulitic appearance of altered pseudotachylyte at different scales.**







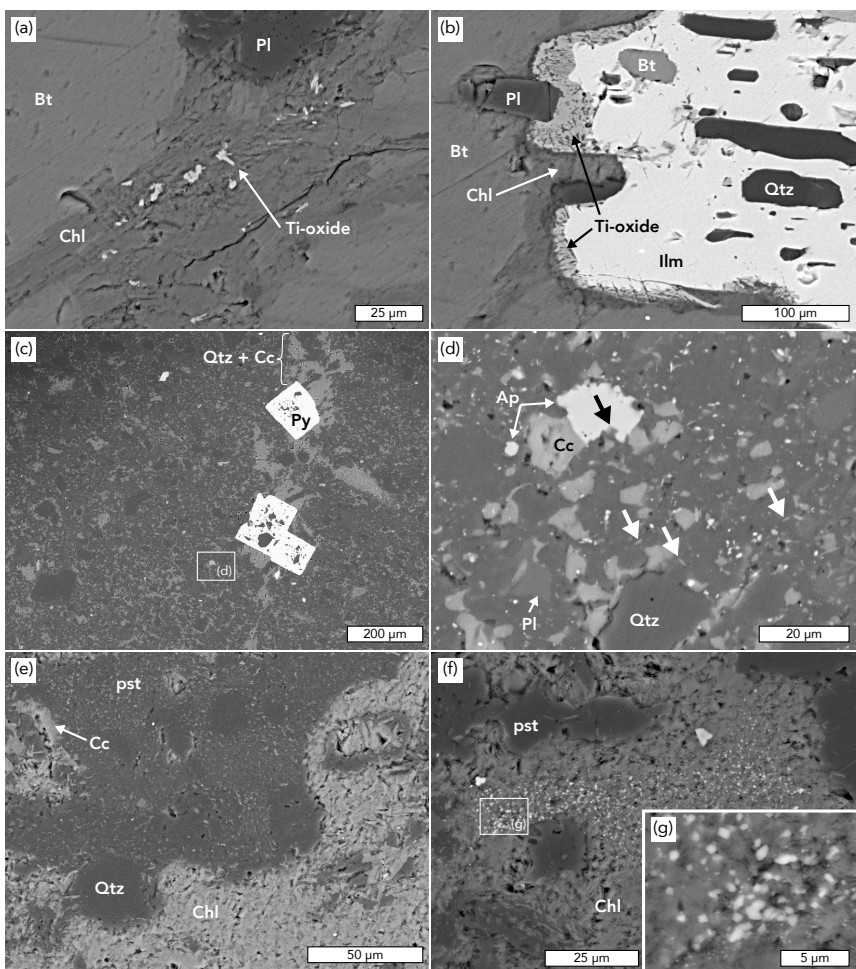

**Figure 5. Back-scattered electron images of selected microscopic observations. Mineral abbreviations after Whitney and Evans (2010) (a) Breakdown of biotite to chlorite + Ti-oxide + unidentified K-phase. (b) Reaction rim around ilmenite inclusion in biotite: ilmenite + biotite = Ti-oxide + chlorite + unidentified K-phase. (c) Pseudotachylyte with vertical, crosscutting vein with calcite +**

**quartz + pyrite. (d) Microstructure of pseudotachylyte. Calcite (light gray) and unknown, needle-shaped (white arrows) microcrystallites, and bright, sub-μm-sized oxide/sulfide droplets are dispersed in cryptocrystalline or amorphous matrix. Note embayed edges (black arrow) of an apatite survivor clast versus a much smaller, euhedral apatite crystal further left. (e) Meandering alteration front between chloritized and pristine pseudotachylyte. No preferred orientation of chlorite. Note, in contrast, the fan-like growth of microcrystallites in the matrix. (f-g) Micrometer-sized crystals of Ti-oxide (determined by EDS)**

**finely dispersed in chlorite which replaced cryptocrystalline/amorphous pseudotachylyte (pst).**





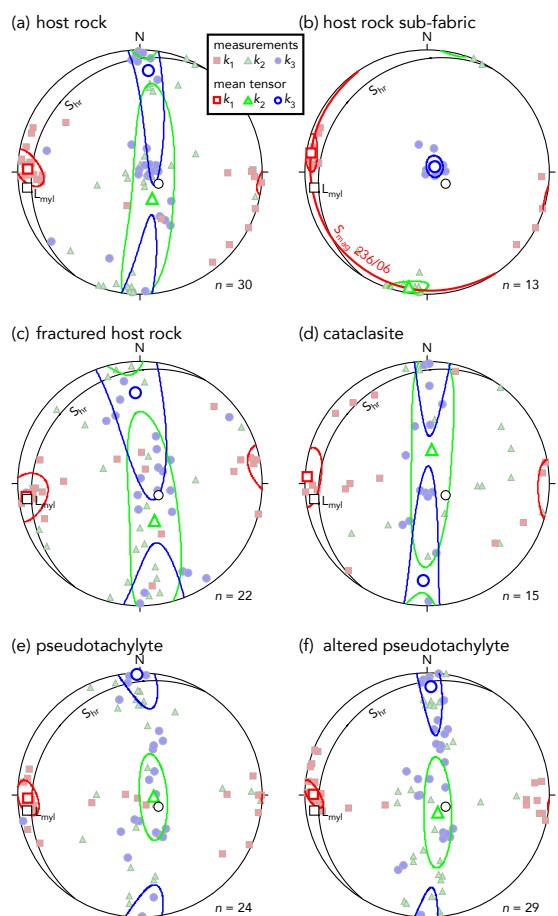

**Figure 6. (a–f) Lower-hemisphere, equal-area plots for principal axes of magnetic anisotropy in different rock types. Comments about data presentation: (a) The measurement for specimen AB15-75 was excluded because it was considered as outlier due to its high $k_m$ (cf. Table 1). (e) All data for specimens containing ≥50% pseudotachylyte was plotted. (f) All data for specimens with containing ≥50% altered pseudotachylyte was plotted.**






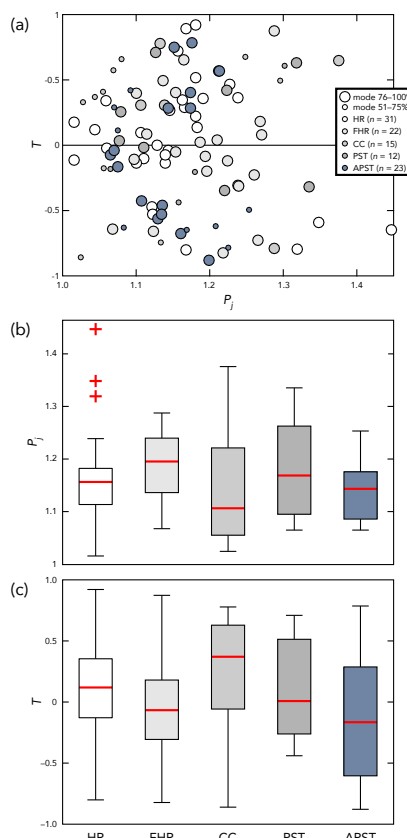

**Figure 7. Variation of anisotropy degree ($P_j$) and shape ($T$) parameters for AMS of different rock types. None of the rock type is**
**distinct from the others based on these parameters (a). (b–c) Box-and-whisker plots for $P_j$ and $T$. HR − host rock, FHR − fractured host rock, CC − cataclasite, PST − pseudotachylyte, APST − altered pseudotachylyte.**



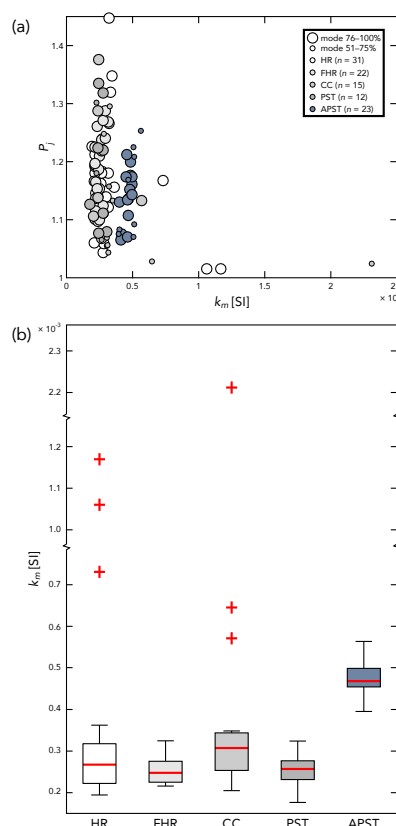

**Figure 8. (a) Normalized mean susceptibility ($k_m$) versus degree of anisotropy ($P_j$). The mean susceptibility of altered**
**pseudotachylyte specimens is about twice that of the other rock type specimens (b). For abbreviations, see Figure 7.**





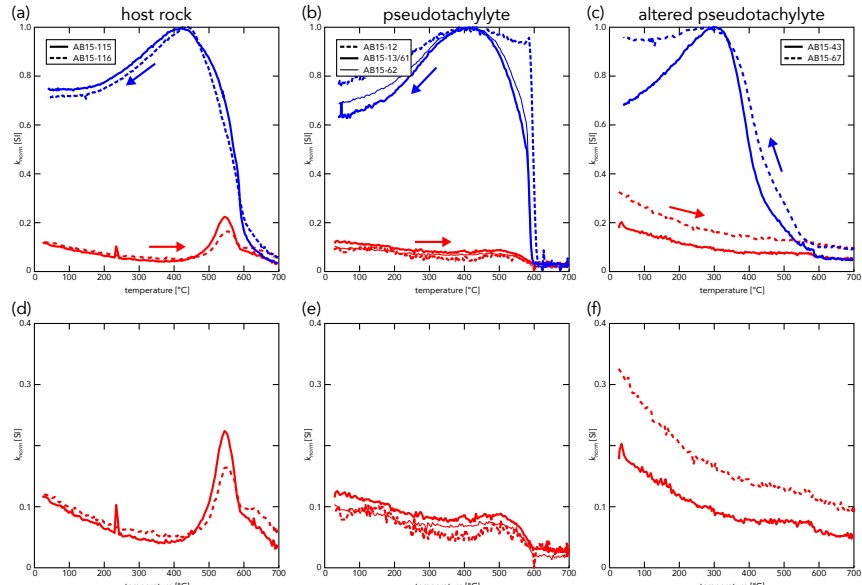

**Figure 9. (a–c) Thermomagnetic curves for host rock, pristine and altered pseudotachylyte during heating from room temperature to 700 °C (red curves) and cooling back to room temperature (blue curves). Susceptibility measurements ($k_{norm}$) were normalized based on the highest value of each sample during the experiment.**






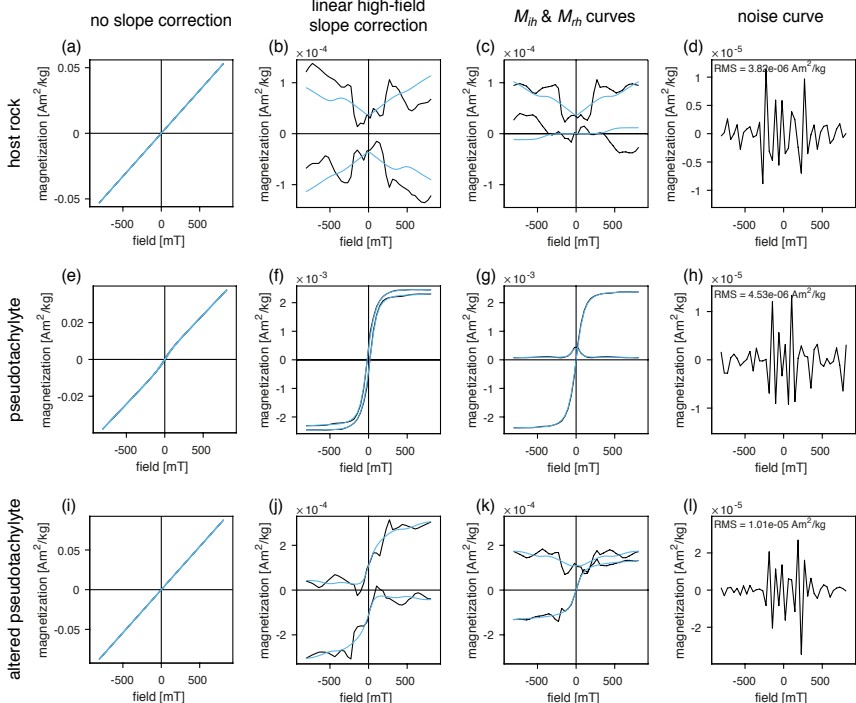

**Figure 10. Hysteresis results for selected host rock, pseudotachylyte and altered pseudotachylyte specimens. Dominant paramagnetic behavior of all samples is displayed in raw measurement curves (left column).**




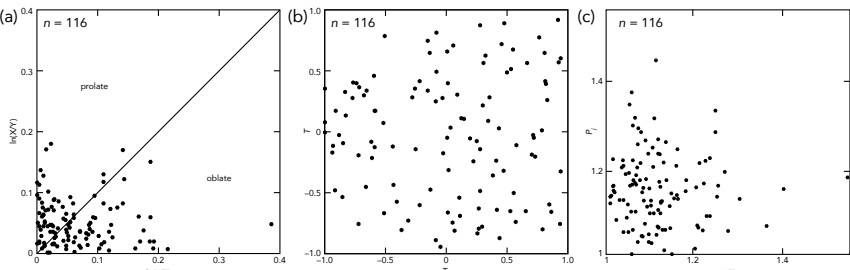

**Figure 11. Comparison of shape parameters for specimen cubes and AMS ellipsoids. (a) Flinn diagram of specimen dimensions shows that cubes are not perfectly isometric. (b–c) Specimen shape does not seem to exert an influence on either shape or degree of magnetic anisotropy.**


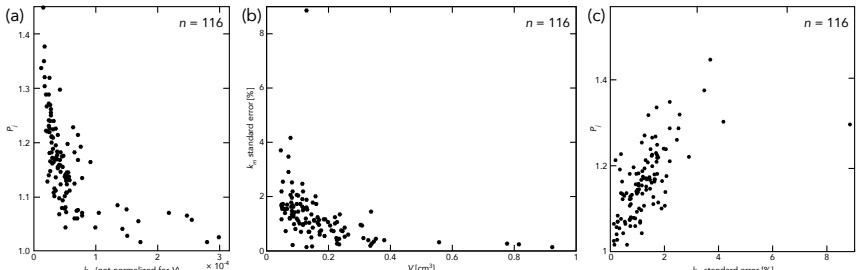

**Figure 12. Influence of specimen size on the degree of magnetic anisotropy. (a) Without normalization for specimen volume, anisotropy degree ($P_j$) decreases with increasing mean susceptibility ($k_m$). (b) Positive correlation between analytical error and $P_j$.**



**Table 1. Anisotropy of magnetic susceptibility data for host rock and different fault rock types.**

| Specimen ID | $k_m$ (raw) ×10⁻⁶ [SI] | $k_m$ (norm.) ×10⁻⁶ [SI] | std. err. [%] | $P_j$ | $T$ | $k_1$ declination [°] | $k_1$ inclination [°] | $k_2$ declination [°] | $k_2$ inclination [°] | $k_3$ declination [°] | $k_3$ inclination [°] | F test | F12 test | F23 test |
|---|---|---|---|---|---|---|---|---|---|---|---|---|---|---|
| colspan | | | | | | | | | | | | | | |

Host rock (n = 31). Median $k_m$ = 266 × 10⁻⁶ SI (all data); mean $k_m$ = 332 ± 230 × 10⁻⁶ SI (all data); mean $k_m$ = 262 ± 46 × 10⁻⁶ SI (without outliers)

mode 76–100%

| Specimen ID | $k_m$ (raw) | $k_m$ (norm.) | std. err. | $P_j$ | $T$ | $k_1$ dec | $k_1$ inc | $k_2$ dec | $k_2$ inc | $k_3$ dec | $k_3$ inc | F test | F12 test | F23 test |
|---|---|---|---|---|---|---|---|---|---|---|---|---|---|---|
| AB15-35 | 17 | 331 | 2.5 | 1.32 | -0.79 | 266 | 6 | 358 | 20 | 159 | 69 | 18.0 | 29.7 | 0.3 |
| AB15-36 | 17 | 346 | 2.2 | 1.35 | -0.59 | 266 | 3 | 357 | 32 | 172 | 57 | 28.5 | 39.0 | 1.6 |
| AB15-37 | 15 | 324 | 3.7 | 1.45 | -0.65 | 265 | 6 | 360 | 36 | 166 | 53 | 15.2 | 21.8 | 0.6 |
| AB15-38 | 35 | 730 | 1.6 | 1.17 | -0.80 | 269 | 0 | 359 | 30 | 179 | 60 | 13.8 | 21.4 | 0.2 |
| AB15-72 | 43 | 317 | 0.7 | 1.12 | -0.47 | 301 | 16 | 66 | 64 | 205 | 20 | 29.5 | 38.5 | 4.7 |
| AB15-73 | 32 | 271 | 1.5 | 1.24 | 0.36 | 270 | 15 | 100 | 75 | 0 | 3 | 29.2 | 7.1 | 29.3 |
| AB15-74 | 51 | 277 | 1.2 | 1.04 | 0.12 | 223 | 58 | 103 | 17 | 4 | 26 | 1.6 | 0.9 | 1.3 |
| AB15-75 | 172 | 1169 | 0.2 | 1.02 | 0.11 | 177 | 59 | 17 | 30 | 282 | 9 | 12.4 | 8.3 | 5.7 |
| AB15-76 | 70 | 285 | 0.7 | 1.06 | 0.34 | 131 | 17 | 223 | 5 | 329 | 72 | 8.3 | 1.7 | 8.2 |
| AB15-96 | 51 | 302 | 1.0 | 1.14 | -0.13 | 89 | 22 | 257 | 67 | 357 | 4 | 27.0 | 21.4 | 11.5 |
| AB15-97 | 37 | 271 | 1.0 | 1.18 | 0.14 | 90 | 24 | 266 | 66 | 359 | 1 | 37.2 | 17.3 | 27.4 |
| AB15-98 | 43 | 316 | 0.7 | 1.18 | 0.22 | 86 | 24 | 229 | 61 | 349 | 15 | 74.9 | 29.0 | 59.8 |
| AB15-99 | 36 | 245 | 1.3 | 1.17 | 0.89 | 88 | 7 | 236 | 82 | 358 | 4 | 18.9 | 0.1 | 33.1 |
| AB15-100 | 43 | 362 | 0.9 | 1.16 | 0.72 | 91 | 43 | 256 | 46 | 354 | 7 | 32.9 | 1.1 | 46.1 |
| AB15-101 | 29 | 266 | 1.7 | 1.18 | 0.92 | 268 | 7 | 66 | 82 | 178 | 3 | 13.2 | 0.0 | 23.0 |
| AB15-102 | 55 | 234 | 1.3 | 1.14 | 0.04 | 95 | 14 | 286 | 76 | 186 | 3 | 15.0 | 10.7 | 8.1 |
| AB15-103 | 57 | 275 | 0.8 | 1.11 | -0.10 | 95 | 11 | 236 | 76 | 4 | 9 | 27.9 | 22.0 | 13.1 |
| AB15-104 | 52 | 240 | 0.7 | 1.13 | 0.00 | 95 | 1 | 197 | 86 | 5 | 4 | 39.1 | 25.9 | 23.1 |
| AB15-105 | 53 | 227 | 0.4 | 1.12 | 0.53 | 89 | 14 | 234 | 73 | 356 | 9 | 101.8 | 135.7 | 11.5 |
| AB15-106 | 43 | 219 | 1.1 | 1.15 | 0.26 | 94 | 8 | 278 | 82 | 184 | 1 | 23.7 | 8.6 | 22.2 |
| AB15-107 | 53 | 242 | 1.2 | 1.11 | 0.10 | 94 | 3 | 284 | 87 | 184 | 0 | 9.8 | 5.3 | 7.1 |
| AB15-108 | 52 | 236 | 1.0 | 1.14 | 0.07 | 88 | 11 | 225 | 75 | 356 | 10 | 24.4 | 18.3 | 11.8 |
| AB15-109 | 50 | 218 | 1.1 | 1.10 | -0.13 | 91 | 6 | 243 | 83 | 1 | 3 | 12.3 | 10.2 | 5.4 |
| AB15-110 | 280 | 1060 | 0.6 | 1.02 | 0.18 | 143 | 14 | 312 | 76 | 52 | 3 | 0.7 | 0.4 | 0.8 |
| AB15-111 | 65 | 212 | 0.9 | 1.06 | 0.02 | 113 | 7 | 22 | 6 | 251 | 81 | 5.7 | 3.3 | 3.2 |
| AB15-112 | 66 | 216 | 1.0 | 1.18 | 0.52 | 108 | 14 | 17 | 3 | 276 | 76 | 42.4 | 5.8 | 52.9 |
| AB15-113 | 71 | 210 | 1.5 | 1.17 | 0.29 | 118 | 13 | 26 | 9 | 263 | 75 | 15.5 | 4.3 | 14.3 |
| AB15-114 | 63 | 193 | 0.4 | 1.23 | 0.46 | 110 | 8 | 19 | 3 | 269 | 82 | 368.6 | 62.0 | 423.9 |
| AB15-115 | 72 | 213 | 0.2 | 1.21 | 0.57 | 111 | 7 | 20 | 4 | 263 | 82 | 1412.3 | 145.8 | 1812.5 |
| AB15-116 | 75 | 217 | 0.4 | 1.19 | 0.36 | 108 | 4 | 18 | 7 | 228 | 82 | 342.4 | 84.7 | 347.0 |
| AB15-117 | 92 | 261 | 0.4 | 1.16 | 0.34 | 110 | 7 | 19 | 5 | 257 | 82 | 171.3 | 42.8 | 171.8 |

Fractured host rock (n = 22). Median $k_m$ = 247 × 10⁻⁶ SI (all data); mean $k_m$ = 254 ± 32 × 10⁻⁶ SI (all data)

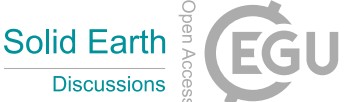

**mode 76–100%**

| | | | | | | | | | | | | | | |
|---|---|---|---|---|---|---|---|---|---|---|---|---|---|---|
| AB15-01 | 27 | 234 | 2.5 | 1.26 | -0.23 | 105 | 14 | 15 | 0 | 283 | 76 | 12.9 | 12.3 | 4.1 |
| AB15-02 | 29 | 247 | 2.2 | 1.21 | 0.04 | 85 | 3 | 355 | 5 | 205 | 85 | 11.2 | 7.1 | 6.9 |
| AB15-03 | 36 | 215 | 1.3 | 1.19 | 0.08 | 82 | 0 | 172 | 11 | 350 | 79 | 29.5 | 23.0 | 13.6 |
| AB15-04 | 47 | 247 | 1.2 | 1.15 | 0.05 | 268 | 2 | 177 | 8 | 8 | 82 | 21.9 | 16.3 | 11.4 |
| AB15-05 | 31 | 242 | 1.2 | 1.19 | 0.02 | 251 | 5 | 341 | 7 | 125 | 82 | 28.6 | 17.8 | 17.3 |
| AB15-14 | 35 | 218 | 1.7 | 1.22 | -0.12 | 88 | 1 | 178 | 32 | 357 | 58 | 19.6 | 15.7 | 6.6 |
| AB15-15 | 50 | 222 | 0.4 | 1.14 | 0.49 | 86 | 0 | 176 | 23 | 355 | 67 | 138.9 | 23.4 | 159.7 |
| AB15-16 | 54 | 215 | 0.9 | 1.20 | -0.20 | 283 | 3 | 192 | 17 | 24 | 73 | 59.3 | 54.8 | 20.0 |
| AB15-34 | 21 | 324 | 1.6 | 1.27 | 0.73 | 265 | 7 | 358 | 19 | 157 | 70 | 34.1 | 54.4 | 1.0 |
| AB15-39 | 26 | 224 | 1.9 | 1.16 | 0.65 | 40 | 23 | 310 | 0 | 220 | 67 | 8.2 | 0.5 | 11.3 |
| AB15-46 | 26 | 265 | 2.1 | 1.24 | 0.31 | 76 | 21 | 191 | 47 | 330 | 35 | 14.5 | 15.3 | 2.8 |
| AB15-47 | 24 | 294 | 2.0 | 1.24 | 0.31 | 76 | 30 | 196 | 41 | 323 | 34 | 14.9 | 16.2 | 3.1 |
| AB15-54 | 27 | 315 | 1.7 | 1.27 | 0.18 | 80 | 12 | 209 | 72 | 347 | 14 | 28.0 | 13.5 | 21.0 |
| AB15-55 | 24 | 276 | 1.3 | 1.27 | 0.08 | 80 | 20 | 202 | 56 | 340 | 26 | 46.9 | 27.8 | 25.9 |
| AB15-56 | 66 | 274 | 0.8 | 1.12 | -0.66 | 78 | 17 | 347 | 4 | 244 | 73 | 33.8 | 50.9 | 2.0 |
| AB15-68 | 38 | 231 | 1.0 | 1.15 | 0.40 | 82 | 37 | 284 | 51 | 180 | 11 | 26.1 | 4.8 | 27.2 |
| AB15-69 | 34 | 224 | 1.8 | 1.11 | 0.09 | 299 | 57 | 60 | 18 | 160 | 27 | 4.5 | 2.5 | 3.0 |
| AB15-70 | 40 | 239 | 1.1 | 1.10 | -0.11 | 329 | 66 | 65 | 3 | 156 | 24 | 8.1 | 6.8 | 3.6 |
| AB15-86 | 21 | 269 | 1.8 | 1.22 | 0.82 | 313 | 76 | 56 | 3 | 147 | 14 | 19.0 | 29.8 | 0.2 |
| AB15-87 | 24 | 292 | 2.2 | 1.29 | 0.87 | 271 | 24 | 87 | 66 | 181 | 2 | 18.5 | 0.1 | 29.3 |
| AB15-94 | 41 | 268 | 1.1 | 1.07 | -0.64 | 176 | 56 | 311 | 25 | 51 | 21 | 4.2 | 5.7 | 0.2 |
| AB15-95 | 33 | 251 | 1.9 | 1.10 | 0.40 | 178 | 18 | 335 | 71 | 85 | 7 | 3.7 | 0.8 | 4.1 |

**Cataclasite ($n = 15$). Median $k_m = 306 \times 10^{-6}$ SI (all data); mean $k_m = 462 \pm 526 \times 10^{-6}$ SI (all data); mean $k_m = 284 \pm 44 \times 10^{-6}$ SI (without outliers)**

**mode 76–100%**

| | | | | | | | | | | | | | | |
|---|---|---|---|---|---|---|---|---|---|---|---|---|---|---|
| AB15-58 | 19 | 238 | 2.5 | 1.29 | 0.79 | 85 | 20 | 191 | 38 | 333 | 46 | 15.3 | 23.5 | 0.2 |
| AB15-59 | 24 | 237 | 2.0 | 1.14 | 0.31 | 326 | 10 | 65 | 40 | 225 | 49 | 4.5 | 1.4 | 4.3 |
| AB15-60 | 32 | 204 | 2.0 | 1.11 | 0.30 | 316 | 2 | 48 | 36 | 224 | 54 | 3.0 | 0.8 | 2.7 |
| AB15-78 | 53 | 570 | 0.8 | 1.13 | 0.78 | 274 | 44 | 86 | 45 | 180 | 4 | 34.9 | 0.7 | 51.2 |
| AB15-79 | 17 | 243 | 3.5 | 1.38 | 0.65 | 270 | 23 | 96 | 67 | 1 | 2 | 11.5 | 0.8 | 15.1 |

**mode 51–75%**

| | | | | | | | | | | | | | | |
|---|---|---|---|---|---|---|---|---|---|---|---|---|---|---|
| AB15-10 | 42 | 329 | 8.8 | 1.30 | 0.49 | 127 | 5 | 36 | 11 | 241 | 78 | 1.2 | 0.1 | 1.3 |
| AB15-17 | 299 | 2312 | 0.1 | 1.02 | 0.86 | 87 | 3 | 185 | 69 | 356 | 21 | 51.0 | 83.8 | 0.4 |
| AB15-18 | 78 | 348 | 0.5 | 1.13 | -0.75 | 89 | 7 | 352 | 47 | 185 | 42 | 113.2 | 164.0 | 2.3 |
| AB15-19 | 77 | 306 | 0.5 | 1.07 | 0.57 | 12 | 9 | 105 | 17 | 254 | 71 | 22.8 | 2.3 | 30.3 |
| AB15-20 | 168 | 302 | 0.3 | 1.05 | 0.41 | 79 | 16 | 348 | 3 | 246 | 74 | 38.7 | 8.2 | 44.9 |
| AB15-21 | 99 | 315 | 0.5 | 1.04 | 0.37 | 42 | 5 | 135 | 33 | 305 | 56 | 9.6 | 1.9 | 9.5 |
| AB15-40 | 255 | 313 | 0.2 | 1.06 | -0.18 | 86 | 14 | 354 | 7 | 237 | 74 | 75.5 | 66.3 | 30.2 |





| | | | | | | | | | | | | | | |
|---|---|---|---|---|---|---|---|---|---|---|---|---|---|---|
| AB15-50 | 151 | 644 | 0.2 | 1.03 | 0.33 | 137 | 37 | 239 | 15 | 346 | 49 | 13.4 | 3.5 | 14.6 |
| AB15-51 | 51 | 288 | 0.7 | 1.08 | 0.66 | 114 | 29 | 232 | 40 | 1 | 36 | 12.3 | 0.8 | 16.5 |
| AB15-77 | 27 | 280 | 1.7 | 1.25 | 0.68 | 266 | 20 | 97 | 70 | 358 | 4 | 22.9 | 1.2 | 31.8 |

**Pseudotachylyte ($n$ = 12). Median $k_m$ = 256 × 10⁻⁶ SI (all data); mean $k_m$ = 256 ± 40 × 10⁻⁶ SI (all data)**

mode 76–100%

| | | | | | | | | | | | | | | |
|---|---|---|---|---|---|---|---|---|---|---|---|---|---|---|
| AB15-12 | 19 | 274 | 2.9 | 1.22 | -0.35 | 98 | 6 | 190 | 15 | 346 | 74 | 7.1 | 7.8 | 1.4 |
| AB15-13 | 12 | 241 | 1.7 | 1.34 | -0.32 | 82 | 5 | 207 | 82 | 352 | 7 | 43.6 | 49.6 | 9.9 |
| AB15-49 | 45 | 304 | 0.9 | 1.08 | 0.25 | 129 | 29 | 237 | 28 | 2 | 47 | 8.5 | 2.7 | 7.6 |
| AB15-61 | 36 | 276 | 1.6 | 1.11 | 0.02 | 115 | 15 | 16 | 32 | 226 | 54 | 5.4 | 3.5 | 2.7 |
| AB15-62 | 50 | 244 | 0.9 | 1.08 | 0.04 | 99 | 16 | 353 | 44 | 204 | 42 | 8.3 | 5.0 | 4.1 |
| AB15-80 | 25 | 275 | 1.4 | 1.32 | 0.63 | 263 | 43 | 86 | 47 | 354 | 1 | 52.3 | 3.4 | 67.0 |
| AB15-89 | 22 | 175 | 1.6 | 1.13 | 0.71 | 84 | 7 | 334 | 71 | 177 | 18 | 7.6 | 0.4 | 11.3 |
| AB15-90 | 31 | 236 | 1.5 | 1.22 | 0.42 | 89 | 43 | 269 | 47 | 179 | 0 | 24.2 | 4.0 | 25.4 |

mode 51–75%

| | | | | | | | | | | | | | | |
|---|---|---|---|---|---|---|---|---|---|---|---|---|---|---|
| AB15-09 | 41 | 323 | 1.0 | 1.16 | -0.44 | 96 | 9 | 188 | 9 | 323 | 78 | 30.5 | 37.5 | 4.9 |
| AB15-42 | 248 | 268 | 0.1 | 1.06 | 0.18 | 90 | 18 | 355 | 16 | 226 | 66 | 310.3 | 275.9 | 116.8 |
| AB15-82 | 18 | 224 | 4.2 | 1.30 | 0.61 | 261 | 74 | 93 | 15 | 2 | 3 | 5.6 | 0.6 | 7.7 |
| AB15-88 | 26 | 225 | 1.6 | 1.18 | -0.21 | 234 | 78 | 346 | 5 | 77 | 11 | 15.4 | 14.3 | 5.1 |

**Altered pseudotachylyte ($n$ = 23). Median $k_m$ = 468 × 10⁻⁶ SI (all data); mean $k_m$ = 469 ± 43 × 10⁻⁶ SI (all data)**

mode 76–100%

| | | | | | | | | | | | | | | |
|---|---|---|---|---|---|---|---|---|---|---|---|---|---|---|
| AB15-26 | 29 | 498 | 1.5 | 1.16 | -0.68 | 271 | 0 | 1 | 26 | 180 | 64 | 14.6 | 21.1 | 0.6 |
| AB15-27 | 29 | 485 | 1.6 | 1.20 | -0.88 | 86 | 0 | 176 | 22 | 356 | 68 | 21.2 | 35.4 | 0.1 |
| AB15-43 | 58 | 402 | 0.5 | 1.13 | 0.56 | 105 | 11 | 205 | 42 | 4 | 46 | 80.8 | 102.5 | 5.8 |
| AB15-48 | 55 | 467 | 1.0 | 1.14 | 0.46 | 104 | 11 | 203 | 39 | 0 | 49 | 23.8 | 27.5 | 2.6 |
| AB15-63 | 35 | 464 | 1.2 | 1.13 | 0.53 | 96 | 6 | 199 | 65 | 4 | 25 | 17.0 | 22.1 | 1.6 |
| AB15-64 | 52 | 468 | 0.7 | 1.11 | -0.42 | 97 | 12 | 200 | 49 | 357 | 39 | 30.2 | 33.7 | 3.9 |
| AB15-65 | 71 | 415 | 0.7 | 1.08 | 0.17 | 100 | 18 | 9 | 3 | 270 | 71 | 14.9 | 12.9 | 6.3 |
| AB15-66 | 78 | 414 | 0.6 | 1.06 | 0.07 | 83 | 20 | 173 | 2 | 268 | 70 | 18.2 | 13.9 | 9.8 |
| AB15-67 | 105 | 460 | 0.3 | 1.07 | 0.04 | 100 | 22 | 193 | 6 | 298 | 67 | 90.3 | 63.2 | 51.0 |
| AB15-83 | 39 | 478 | 1.3 | 1.15 | 0.75 | 260 | 48 | 92 | 42 | 357 | 6 | 15.9 | 0.4 | 23.1 |
| AB15-84 | 37 | 483 | 2.1 | 1.18 | 0.79 | 262 | 21 | 104 | 67 | 355 | 8 | 8.6 | 0.2 | 12.9 |
| AB15-85 | 38 | 457 | 1.3 | 1.21 | 0.57 | 91 | 12 | 251 | 78 | 0 | 4 | 30.8 | 3.8 | 41.2 |
| AB15-91 | 53 | 497 | 1.0 | 1.14 | 0.28 | 93 | 34 | 260 | 56 | 359 | 6 | 25.1 | 7.0 | 22.9 |
| AB15-92 | 48 | 497 | 1.2 | 1.17 | 0.40 | 93 | 25 | 276 | 65 | 183 | 1 | 25.5 | 5.4 | 27.8 |
| AB15-93 | 46 | 453 | 1.5 | 1.17 | 0.28 | 96 | 33 | 273 | 57 | 5 | 2 | 17.0 | 4.9 | 15.6 |

mode 51–75%

| | | | | | | | | | | | | | | |
|---|---|---|---|---|---|---|---|---|---|---|---|---|---|---|
| AB15-23 | 38 | 466 | 1.6 | 1.17 | -0.65 | 87 | 5 | 183 | 44 | 352 | 45 | 14.3 | 19.4 | 0.6 |





| | | | | | | | | | | | | | |
|---|---|---|---|---|---|---|---|---|---|---|---|---|---|
| AB15-28 | 31 | 512 | 1.4 | 1.21 | -0.62 | 88 | 5 | 181 | 34 | 350 | 55 | 28.7 | 38.8 | 1.4 |
| AB15-29 | 27 | 563 | 1.2 | 1.25 | -0.49 | 85 | 1 | 175 | 5 | 341 | 85 | 58.8 | 79.0 | 7.3 |
| AB15-30 | 26 | 506 | 1.5 | 1.23 | -0.78 | 88 | 3 | 357 | 29 | 184 | 61 | 27.4 | 42.1 | 0.4 |
| AB15-44 | 136 | 399 | 0.3 | 1.08 | -0.63 | 102 | 15 | 194 | 7 | 311 | 74 | 122.1 | 175.3 | 8.5 |
| AB15-45 | 150 | 394 | 0.4 | 1.08 | 0.11 | 83 | 26 | 182 | 17 | 301 | 58 | 45.1 | 24.9 | 34.5 |
| AB15-52 | 46 | 515 | 1.0 | 1.09 | 0.42 | 112 | 9 | 9 | 55 | 208 | 33 | 9.2 | 2.1 | 9.7 |
| AB15-53 | 52 | 507 | 1.1 | 1.07 | 0.29 | 112 | 12 | 2 | 59 | 209 | 28 | 4.3 | 1.5 | 4.0 |

**Mixed specimens with either more than two rock types or 50/50 mode**

50% fractured host rock, 25% pseudotachylite, 25% altered pseudotachylite

| | | | | | | | | | | | | | |
|---|---|---|---|---|---|---|---|---|---|---|---|---|---|
| AB15-71 | 58 | 254 | 0.9 | 1.06 | 0.54 | 52 | 81 | 267 | 8 | 177 | 5 | 6.2 | 0.7 | 8.2 |

50% cataclasite, 50% pristine pseudotachylyte ($n = 6$). Median $k_m = 325 \times 10^{-6}$ SI (all data); mean $k_m = 666 \pm 592 \times 10^{-6}$ SI (all data)

| | | | | | | | | | | | | | |
|---|---|---|---|---|---|---|---|---|---|---|---|---|---|
| AB15-07 | 42 | 278 | 1.1 | 1.17 | -0.58 | 103 | 6 | 197 | 36 | 6 | 54 | 26.4 | 34.5 | 1.7 |
| AB15-08 | 56 | 327 | 0.6 | 1.14 | 0.18 | 65 | 3 | 155 | 2 | 273 | 87 | 65.1 | 24.9 | 51.4 |
| AB15-11 | 26 | 323 | 1.5 | 1.15 | -0.24 | 73 | 9 | 339 | 24 | 182 | 64 | 12.4 | 11.1 | 3.4 |
| AB15-41 | 218 | 281 | 0.3 | 1.07 | 0.17 | 90 | 15 | 356 | 17 | 219 | 67 | 87.9 | 75.4 | 33.5 |
| AB15-57 | 69 | 1108 | 0.7 | 1.07 | 0.47 | 80 | 20 | 215 | 62 | 343 | 18 | 13.9 | 17.1 | 1.9 |
| AB15-81 | 142 | 1680 | 0.2 | 1.04 | 0.82 | 271 | 66 | 89 | 24 | 179 | 1 | 46.6 | 0.8 | 75.6 |

50% pristine pseudotachylyte, 50% altered pseudotachylyte ($n = 6$). Median $k_m = 448 \times 10^{-6}$ SI (all data); mean $k_m = 427 \pm 57 \times 10^{-6}$ SI (all data)

| | | | | | | | | | | | | | |
|---|---|---|---|---|---|---|---|---|---|---|---|---|---|
| AB15-22 | 30 | 348 | 1.3 | 1.17 | -0.93 | 87 | 4 | 356 | 25 | 185 | 65 | 22.0 | 37.3 | 0.0 |
| AB15-24 | 35 | 460 | 1.2 | 1.17 | 0.72 | 87 | 3 | 179 | 31 | 352 | 59 | 26.0 | 38.0 | 0.7 |
| AB15-25 | 31 | 369 | 1.6 | 1.16 | -0.73 | 88 | 2 | 179 | 33 | 355 | 57 | 13.4 | 19.6 | 0.3 |
| AB15-31 | 25 | 460 | 1.7 | 1.22 | -0.77 | 83 | 1 | 352 | 27 | 176 | 63 | 20.3 | 31.2 | 0.4 |
| AB15-32 | 25 | 493 | 1.7 | 1.23 | -0.64 | 84 | 0 | 354 | 28 | 174 | 62 | 22.9 | 32.3 | 1.1 |
| AB15-33 | 29 | 435 | 1.7 | 1.21 | -0.74 | 262 | 1 | 352 | 23 | 170 | 67 | 18.8 | 29.0 | 0.5 |

 

**Table 2. Magnetic hysteresis parameters for processed hysteresis loops (automatic drift correction and linear high-field slope**
**correction, with a cut-off field of 567 mT). Hysteresis data has been mass normalized.**

| Specimen ID | mass [mg] | $M_s$ [Am$^2$/kg] | $M_{rs}$ [Am$^2$/kg] | $B_c$ [mT] | $X$ [m$^3$/kg] | Notes |
|---|---|---|---|---|---|---|
| *Host rock* | | | | | | |
| AB15-75 | 376 | 1.40E-04 | 2.06E-05 | — | 9.22E-08 | very open loop |
| AB15-99 | 378 | 4.68E-04 | 1.78E-04 | 36.2 | 8.89E-08 | semi-closed loop, closing loop |
| AB15-115 | 881 | 9.41E-05 | 3.35E-05 | — | 8.25E-08 | very open loop |
| AB15-116 | 919 | 1.01E-04 | 3.00E-05 | — | 7.11E-08 | very open loop |
| *Fractured host rock (*5 Vol-% pseudotachylyte)* | | | | | | |
| AB15-04 | 450 | 3.49E-04 | 1.92E-04 | 39.6 | 8.23E-08 | open loop |
| AB15-16 | 624 | 4.01E-04 | 2.30E-04 | 50.7 | 8.10E-08 | open loop |
| AB15-56* | 587 | 1.46E-03 | 1.51E-04 | 12.5 | 9.12E-08 | open loop, closing loop |
| *Cataclasite (and *40 Vol-% or †10 Vol-% pristine pseudotachylyte)* | | | | | | |
| AB15-17* | 315 | 2.43E-03 | 3.30E-04 | 10.4 | 5.99E-08 | open loop |
| AB15-20* | 1419 | 2.00E-03 | 2.16E-04 | 10.8 | 8.21E-08 | open loop |
| AB15-60† | 382 | 1.88E-03 | 2.37E-04 | 10.8 | 7.37E-08 | open loop |
| *Pseudotachylyte (and *20–40 Vol-% or †5–10 Vol-% cataclasite)* | | | | | | |
| AB15-12 | 162 | 1.41E-03 | 5.50E-04 | 31.2 | 5.60E-08 | semi-closed loop, closing loop |
| AB15-13 | 112 | 2.01E-03 | 6.23E-04 | 28.0 | 6.30E-08 | semi-closed loop, closing loop |
| AB15-42* | 2366 | 1.10E-03 | 1.49E-04 | 11.2 | 7.54E-08 | open loop |
| AB15-49* | 378 | 1.79E-03 | 1.95E-04 | 10.8 | 7.71E-08 | open loop, closing loop |
| AB15-61† | 306 | 2.31E-03 | 3.00E-04 | 12.0 | 7.20E-08 | semi-closed loop, closing loop |
| AB15-62† | 508 | 2.38E-03 | 4.37E-04 | 17.3 | 5.51E-08 | semi-closed loop, closing loop |
| AB15-89† | 282 | 1.72E-03 | 4.73E-04 | 25.4 | 4.81E-08 | open loop |
| *Altered pseudotachylyte (*and 5 Vol-% pristine pseudotachylyte)* | | | | | | |
| AB15-43 | 337 | 1.57E-04 | 1.12E-04 | 240.0 | 1.36E-07 | very open loop |
| AB15-48 | 285 | 2.17E-04 | 8.48E-05 | — | 1.75E-07 | very open loop |
| AB15-67 | 574 | 1.35E-04 | 1.13E-04 | 54.3 | 1.75E-07 | very open loop |
| AB15-84* | 202 | 2.40E-04 | 1.92E-04 | 108.5 | 1.82E-07 | very open loop |
| AB15-91 | 263 | 2.65E-04 | 7.83E-05 | — | 2.00E-07 | very open loop |