# Peer review of "Magnetic properties of pseudotachylytes from western Jämtland, Central Swedish Caledonides"

_Solid Earth, 2019_

## Referee Comment (RC1) · Anonymous Referee #1 · 9 Sep 2019

In its present shape, the manuscript requires major revisions before being possibly published in Solid Earth.

What is the message of the paper? The objectives are not clear at all.

The structural analysis is confused and the conclusions are neither exciting nor convincing.

Although I am not a specialist of rock magnetism, the same can be said regarding the magnetic analysis (see below).

1/ Comments on structural analysis

As a structural geologist and regular reader of Solid Earth, I am disappointed by the

structural analysis of pseudotachylyte-bearing fault zones presented here. Particularly, shear senses are poorly constrained and explanations are somewhat confusing. Clarification and reformulation are needed.

The pst (pseudotachylyte) macroscale is somewhat disappointing. Photographs are scarce and poorly informative. PST microscale description is confused. Pst microscale description should consist of a description of matrix, newly crystallized minerals, survivor clasts and other specific features (sulfide droplets, flow folds and so on).

Lines 32-34 : why is it important to compare the pst data with kinematic data from post-orogenic extensional faults?

Lines 65-69 : the authors state that ''mylonitic shear sense indicators . . . were not observed''. Can this missing information be found in the literature? How can the authors discuss the evolution of the nappe complex (with in-sequence and out-of-sequence thrusting and so on) if the early kinematics are unknown?

Lines 72 and following. Fractured fault rocks are not fault rocks, they are usually referred to as ''fractured host rock'' or fractured protolith''. My feeling is that altered pst should not be distinguished from unaltered pst in the fault rock catalog, since formation is the same for both psts.

Lines 79-80 : ''Deflection. . . respectively'' : this sentence is confusing. No photograph of the structures used for establishing the sense of shear. What injection vein asymmetry do the authors refer to?

Lines 81-84 : what are these N-S faults? Do they cross-cut the thrust-sheet bounding faults? More generally, what is the relationship between pst and pst magnetism and those late-stage high-angle normal faults???

Line 160 : what is the meaning of the sentence ''Only . . . pseudotachylyte''?

Lines 165-166 : is calcite a secondary, newly formed mineral? If yes, it is not a survivor clast. Please clarify. It seems that calcite has nothing to do with pst formation. It should

be described separately from psts. And what is the usefulness to describe calcite that has nothing to do with pst as well as with pst magnetic properties???

Line 170 : could it be sanidine?

2/ Comments on magnetism analysis

Given the poor quality of the manuscript, I cannot spend time checking the magnetic side of the analysis presented in the manuscript. Data look scare and poorly understood. Conclusions are not exciting. The authors fail to apply the Ferré et al analysis leading to the reconstruction of focal mechanisms of pst-generating earthquakes.

Line 282: What is the meaning of this sentence ''Fault vein margins...fault zone''. What does ''Seismic faulting in these veins'' mean? The remaining part of this section (lines 283 to 296) is confusing and should be seriously reconsidered.

3/ Phrasing concerns

Line 36 : what is a magmatic assemblage in a pst? I would use ''neoformed'' or newly crystallized'' or something like that since a pst is not exactly a magmatic rock.

Line 37 : delete ''commonly''.

Line 58 : What is a calcareous volcanic rock?

Line 60 : I cannot understand the meaning of ''their strike follow the shape of the synform''. Please reformulate.

Line 66 : the lineation is not carried by biotite and boudinaged amphibole, it is carried by a foliation.

Line 91 : add ''to'' after ''parallel''.

Line 97 : replace ''is'' by ''are''.

Line 146 : I do not like ''Microstructural appearance of host and fault rocks''. Better use more accurate words.

Line 148 : I do not feel confortable with ''porphyroblastic biotite''. Porphyroblasts commonly consist of feldspar, garnet, staurolite and so on. I doutd that biotite can form prophyroblats.

Line 152 : Which cataclastic fault rock do you refer to?

Line 162 : I would replace ''partial melting'' by ''melt corrosion''.

Lines 179-180 : the sentence is unclear and is redundant.

4/ Other issues

A lot of references are cited in the text but miss in the list. Please complete.

Photomicrographs are in some cases poorly legible. Some annotation or drawings could improve legibility. Captions are not helpful and should be reformulated.

For instance, line 450. Caption of Fig. 3 is unclear. ''Figure 3. Macroscopic appearance of a foliation-parallel fault vein exhibiting different kinds of fault rock. For detailed description, see text. Characterization of fault rock types is also based on microscopic observations. The image represents the XZ plane of the ductile finite strain ellipsoid.'' A fault vein consists of pst, not of different kinds of fault rocks. What the ''ductile finite strain ellipsoid''? Why not the brittle one? Psts are brittle structures.

―――――――――――――――――――

---

## Referee Comment (RC2) · Ann Hirt (Referee) · 10 Sep 2019

The authors examine the magnetic properties, including magnetic fabric in small samples taken from rocks from the Köli Nappe complex. Although the original goal of the authors was to use magnetic fabrics to gain a better understanding of kinematics of a ductile-to-brittle shear zone, they could only show that very small samples may not accurately reflect detailed kinematic information within the shear zone. There were a number of studies that tried to use magnetic fabrics in large-scale shear zones in the early 1970's thru 1980's to gain kinematic information. Many of these were unsuccessful, which led to the suggestion that non-homogenous deformation within the shear zone, or in some cases, late stage deformation overprinting any earlier fabrics due to retrograde metamorphism were responsible for the observed magnetic fabrics. It was

not until Ferré and co-workers work that this problem has been looked at by focusing on pseudotachylytes.

There are some interesting points made in this paper, such as the fact that one needs to consider whether a "sample" is representative of a larger volume of rock, or what fabric is one observing if a rock has undergone inhomogeneous deformation or multiple deformation phases. The authors, however, need to better develop these points in the manuscript in order that it makes a significant contribution to the field.

The following are comments are in relationship to the magnetic fabrics, and these can be divided into the directional information, or the degree and shape of the AMS ellipsoids. Note that numerous studies have shown that AMS is very good in reflecting preferred directions of deformation. The degree of anisotropy is also often related to the degree of deformation, but not always, and the shape of the AMS ellipsoid is often poorly constrained or at least the most variable parameter.

Directional data from magnetic fabrics agrees with petrofabric and indicates that the host rocks and rocks from the fault developed in the same strain field. Is there any indication that the petrofabric post-dates the faulting? This is an important question in phyllosilicates carry the AMS and these arise from alteration after the faulting event. Can neo-formation of mica and/or biotite account for the common fabric in all rocks? Could Ti-rich oxide be contributing to the paramagnetic contribution? How homogeneous is the mineralogy in the different categories?

There is a clear relationship between the degree of AMS (Pj) and the "not normalized" mean susceptibility, and the error in mean susceptibility versus sample size, which leads the authors conclude the sample size affects Pj. There are a couple of points that need more clarification. 1) How were the samples measure with the manual 15-position scheme, 3 rotation planes, or "single" rotation scheme?

2) What is the analytical error (km standard error) shown in Fig. 12 c; is it obtained from the SAFYR program?

3) Is the higher P-value related to a weak susceptibility? Normally this is found when mean susceptibility is close to "zero", due to a diamagnetic component of the susceptibility balancing out a para-/ferromagnetic component. In this case is it due to the fact that the small sizes leads to a susceptibility that gets within the accuracy range of the bridge?

4) It appears that the grain size of samples is much smaller that the sample size, but is this really the case, i.e., are there enough grains to reflect the anisotropy of a larger volume? I once tried with a shale/slate sample and a biotite crystal to reduce sample size of a cube to see if this affected the AMS. Although I did not go below a 1-cm edge, I did not see an effect. But in these cases, I had very fine grain sizes with respect to volume or a single crystal.

5) Going back to the point with directional data, how variable is the mineralogy between the different types of samples? How variable is the high-field susceptibility extracted from the magnetization versus field measurements?

6) Shape is often never a good parameter in looking at deformation, and it is not surprising that the shapes are so variable.

Minor comments 1) The authors mention frequency-dependent susceptibility, but do not mention it further in the text. Did they try to measure the AMS at different frequencies, e.g., in the PST APST samples?

2) I am not sure how significant an isolation of a ferromagnetic component is in the host rock or APST. I would put little faith in trying to extract a saturation magnetization. They are surely artefacts and I would not even show. In this case, it would have been better to measure the acquisition of IRM. One could have probably gotten a convincingly significant signal that would allow for comparison.

Line 18: remove hyphen in information

Line 35: 2.1 instead of 1.1

Lines108-109: in the case that T = 1 or -1, then the ellipsoid is rotationally oblate, respectively prolate. The ellipsoid is still oblate for T> 0, and prolate for T < 0.

Line 122: Note that frequency dependence should be detected for particles between ca. 16 to ca. 30 nm for 976 Hz and ca. 15 – ca. 30 nm for 15616 Hz (cf., Hrouda, 2011, GJI).

Line 189: I am not sure what is meant by homogeneous AMS fabrics? Can there be inhomogeneous fabrics?

Note that numerous references within the manuscript or not in the reference list. The authors should go through this carefully. Some reference for generic information are not needed are do not really reflect the authors who originally presented an idea.

Fig. 9: complete figure caption or state that the lower figure shows only the heating curves for a) – c) without labelling.

---

## Author Comment (AC1) · 7 Feb 2020

Dear editor and reviewers,

Here we address the comments raised by the two reviewers of the manuscript. Our answer to a comment is bounded by dashed lines, to make it easier to separate comment and question. Significant changes have been made to the manuscript, in order to attempt clarification of the objective and message of the paper. We have also made a change to the authorship list, whereby Bjarne Almqvist is now listed as lead author and Hagen Bender is the second author. This change in authorship has been approved by all authors of the manuscript.

The title of the paper has changed to: "Magnetic properties of pseudotachylytes from

western Jämtland, central Swedish Caledonides"

Thank you for your consideration. Bjarne Almqvist and Hagen Bender —————————

Anonymous Referee #1 In its present shape, the manuscript requires major revisions before being possibly published in Solid Earth. What is the message of the paper? The objectives are not clear at all. The structural analysis is confused and the conclusions are neither exciting nor convincing.

————————— The objective of the study was originally to obtain detailed kinematic information on faulting that had occurred in the internal part of the Köli nappe in the central Swedish Caledonides. However, the study did not end up with a clear-cut answer in response to the initial goal that was set. This is part of the reason why the message and objective do not appear clear. However, we believe that we make observations of magnetic properties and fabric of the pseudotachylytes that are of general interest and can benefite other researchers that are targeting magnetic fabrics in pseudotachylytes. We have attempted in the revised manuscript to elucidate the objective of the study and make it clearer what the message of the paper is.

In the study of magnetic fabrics there is an inherent challenge in measurement of small samples, which is highlighted. Unfortunately, we cannot do much about the data itself. Despite this issue we would like to stress that we do obtain meaningful magnetic fabrics, which correspond to the structural reference frame (i.e., foliation and lineation). —————————

Although I am not a specialist of rock magnetism, the same can be said regarding the magnetic analysis (see below).

1/ Comments on structural analysis As a structural geologist and regular reader of Solid Earth, I am disappointed by the structural analysis of pseudotachylyte-bearing fault zones presented here. Particularly, shear senses are poorly constrained and explanations are somewhat confusing. Clarification and reformulation are needed. ——

—————

We have tried to accommodate the comments, criticisms and suggestions of the reviewer in order to improve the manuscript. The shear sense is unfortunately not known.
————————

The pst (pseudotachylyte) macroscale is somewhat disappointing. Photographs are scarce and poorly informative. PST microscale description is confused. Pst microscale description should consist of a description of matrix, newly crystallized minerals, survivor clasts and other specific features (sulfide droplets, flow folds and so on). Lines 32-34 : why is it important to compare the pst data with kinematic data from post-orogenic extensional faults?

———————— We have added text to this sentence to indicate that it is of relevance to understand the relationship between the late orogenic stage top W extension and the formation of brittle deformation pseudotachylytes.

———————— Lines 65-69 : the authors state that ''mylonitic shear sense indicators : : : were not observed''. Can this missing information be found in the literature? How can the authors discuss the evolution of the nappe complex (with in-sequence and out-of-sequence thrusting and so on) if the early kinematics are unknown?

———————— We have added text to indicate that shear sense indicators in mylonites have been mapped regionally by Bender et al. (2019) in the lower and middle Köli nappe. In addition, there is a body of work, including Bender et al. (2018) that show dominant top to E shear sense indicators, which prompted the the out-of-sequence thrusting model.

——————

Lines 72 and following. Fractured fault rocks are not fault rocks, they are usually referred to as ''fractured host rock'' or fractured protolith''. My feeling is that altered pst should not be distinguished from unaltered pst in the fault rock catalog, since formation

is the same for both psts.

——————————— We have changed the wording to "fractured host rock". The pseudo-tachylyte rocks have been grouped together now as suggested by the reviewer, but we make distinctions between preserved and altered pseudotachylyte, as this is important for further working with the properties of the pst.

——————————

Lines 79-80 : "Deflection: : : respectively" : this sentence is confusing. No photograph of the structures used for establishing the sense of shear. What injection vein asymmetry do the authors refer to?

——————————— This sentence has been removed in the revised manuscript

——————————

Lines 81-84 : what are these N-S faults? Do they cross-cut the thrust-sheet bounding faults? More generally, what is the relationship between pst and pst magnetism and those late-stage high-angle normal faults???

——————————— We have added text to the end of this paragraph to indicate the likely origin of these N-S striking faults. They are likely late structure that cut across the thrust-sheet bounding faults.

——————————

Line 160 : what is the meaning of the sentence "Only : : : pseudotachylyte"?

——————————— We have removed this sentence and added a sentence in section 3 on the pseudotachylyte terminology. The reason for the original statement is that for the rocks that are investigated it was not initially clear that they were pseudotachylyte.

——————————

Lines 165-166: is calcite a secondary, newly formed mineral? If yes, it is not a survivor

clast. Please clarify. It seems that calcite has nothing to do with pst formation. It should be described separately from psts. And what is the usefulness to describe calcite that has nothing to do with pst as well as with pst magnetic properties???

——————— We have rewritten this sentence and indicate that calcite is most likely a survivor clast. We describe the presence of calcite for completion of the description of the fault rock microstructure and petrography.

——————— Line 170 : could it be sanidine?

——————— We have noted that this could potentially be sanidine or anorthoclase, and made reference to a paper by Lin (1994), where K-feldspar microliths were identified in glassy pseudotachylyte.

———————

2/ Comments on magnetism analysis Given the poor quality of the manuscript, I cannot spend time checking the magnetic side of the analysis presented in the manuscript. Data look scare and poorly understood. Conclusions are not exciting. The authors fail to apply the Ferré et al analysis leading to the reconstruction of focal mechanisms of pst-generating earthquakes.

——————— Yes, in this study we fail to apply the method of Ferré to reconstruct direction and sense of seismic slip. However, we believe we make some interesting observation that may be of use to other researchers working in this topic. We believe that a valid question that can be considered from this study, where more than 100 samples were studied in a detailed, systematic and careful way, is why the method failed to provide an answer on the fault kinematics? Perhaps the obvious answer is that the samples used in the study were not suitable for kinematic analysis (even though the magnetic fabric reflect the petrofabric), and this may be a useful result as well. . .

———————

Line 282: What is the meaning of this sentence ''Fault vein margins: : :fault zone''.

What does ''Seismic faulting in these veins" mean? The remaining part of this section (lines 283 to 296) is confusing and should be seriously reconsidered.

——————— We have rewritten the first two sentences of this paragraph to make the meaning of the sentences clearer (the sentences are used to place the magnetic results in perspective to the structural results). The remainder of the paragraph has been rewritten to improve the clarity of the text.

——————

3/ Phrasing concerns Line 36 : what is a magmatic assemblage in a pst? I would use ''neoformed" or newly crystallized" or something like that since a pst is not exactly a magmatic rock.

——————— We changed the phrasing to "newly crystallized"

——————

Line 37 : delete ''commonly".

——————— done

——————

Line 58 : What is a calcareous volcanic rock?

——————— The word 'calcareous' has been removed

——————

Line 60 : I cannot understand the meaning of ''their strike follow the shape of the synform". Please reformulate.

——————— We have reformulated this sentence and hopefully made it clearer

——————

Line 66 : the lineation is not carried by biotite and boudinaged amphibole, it is carried

by a foliation.

——————— The sentence has been rephrased to indicate that the orientation of biotite and amphibole crystals in the foliation plane show the lineation

——————

Line 91 : add ''to'' after ''parallel''.

——————— done

——————

Line 97 : replace ''is'' by ''are''.

——————— done

——————

Line 146 : I do not like ''Microstructural appearance of host and fault rocks''. Better use more accurate words.

——————— The headline of the section has been changed to "Microstructural description of host and fault rocks"

——————

Line 148 : I do not feel confortable with ''porphyroblastic biotite''. Porphyroblasts commonly consist of feldspar, garnet, staurolite and so on. I doutd that biotite can form prophyroblats.

——————— We have changed the wording to 'Large biotite crystals', in order to avoid the terminology related to porphyroblasts.

——————

Line 152 : Which cataclastic fault rock do you refer to?

———————— It is unclear to us here what the reviewer means about the cataclastic fault rock in this text line. The paragraph itself describes the host rock microstructure and petrography.

————————

Line 162 : I would replace ''partial melting" by ''melt corrosion".

———————— Changed to "melt-assisted corrosion"

————————

Lines 179-180 : the sentence is unclear and is redundant. 4/ Other issues A lot of references are cited in the text but miss in the list. Please complete.

———————— Missing references have been added.

————————

Photomicrographs are in some cases poorly legible. Some annotation or drawings could improve legibility. Captions are not helpful and should be reformulated. For instance, line 450. Caption of Fig. 3 is unclear. ''Figure 3. Macroscopic appearance of a foliation-parallel fault vein exhibiting different kinds of fault rock. For detailed description, see text. Characterization of fault rock types is also based on microscopic observations. The image represents the XZ plane of the ductile finite strain ellipsoid." A fault vein consists of pst, not of different kinds of fault rocks. What the ''ductile finite strain ellipsoid"? Why not the brittle one? Psts are brittle structures.

———————— We have rewritten the figure caption of figure 3, to try to make it clearer. We have also look at other figure captions in order to make them more descriptive and helpful.

———————— Interactive comment on Solid Earth Discuss., https://doi.org/10.5194/se-2019-128, 2019.

---

## Author Comment (AC2) · 7 Feb 2020

Dear editor and reviewers,

Here we address the comments raised by the two reviewers of the manuscript. Our answer to a comment is bounded by dashed lines, to make it easier to separate comment and question. Significant changes have been made to the manuscript, in order to attempt clarification of the objective and message of the paper. We have also made a change to the authorship list, whereby Bjarne Almqvist is now listed as lead author and Hagen Bender is the second author. This change in authorship has been approved by all authors of the manuscript.

The title of the paper has changed to: "Magnetic properties of pseudotachylytes from

**western Jämtland, central Swedish Caledonides"**

Thank you for your consideration. Bjarne Almqvist and Hagen Bender

The authors examine the magnetic properties, including magnetic fabric in small samples taken from rocks from the Köli Nappe complex. Although the original goal of the authors was to use magnetic fabrics to gain a better understanding of kinematics of a ductile-to-brittle shear zone, they could only show that very small samples may not accurately reflect detailed kinematic information within the shear zone. There were a number of studies that tried to use magnetic fabrics in large-scale shear zones in the early 1970's thru 1980's to gain kinematic information. Many of these were unsuccessful, which led to the suggestion that non-homogenous deformation within the shear zone, or in some cases, late stage deformation overprinting any earlier fabrics due to retrograde metamorphism were responsible for the observed magnetic fabrics. It was not until Ferré and co-workers work that this problem has been looked at by focusing on pseudotachylytes. There are some interesting points made in this paper, such as the fact that one needs to consider whether a "sample" is representative of a larger volume of rock, or what fabric is one observing if a rock has undergone inhomogeneous deformation or multiple deformation phases. The authors, however, need to better develop these points in the manuscript in order that it makes a significant contribution to the field.

----------- In the revised manuscript we have tried to develop these points and improve the clarity of the manuscript in general (see also the answers to comments of the first reviewer).

The following are comments are in relationship to the magnetic fabrics, and these can be divided into the directional information, or the degree and shape of the AMS ellipsoids. Note that numerous studies have shown that AMS is very good in reflecting preferred directions of deformation. The degree of anisotropy is also often related to the degree of deformation, but not always, and the shape of the AMS ellipsoid is often Interactive comment

poorly constrained or at least the most variable parameter.

Directional data from magnetic fabrics agrees with petrofabric and indicates that the host rocks and rocks from the fault developed in the same strain field. Is there any indication that the petrofabric post-dates the faulting? This is an important question in phyllosilicates carry the AMS and these arise from alteration after the faulting event.

Can neo-formation of mica and/or biotite account for the common fabric in all rocks? Could Ti-rich oxide be contributing to the paramagnetic contribution? How homogeneous is the mineralogy in the different categories?

There is a clear relationship between the degree of AMS (Pj) and the "not normalized" mean susceptibility, and the error in mean susceptibility versus sample size, which leads the authors conclude the sample size affects Pj. There are a couple of points that need more clarification. 1) How were the samples measure with the manual 15-position scheme, 3 rotation planes, or "single" rotation scheme?

SED
2) What is the analytical error (km standard error) shown in Fig. 12 c; is it obtained from the SAFYR program?

3) Is the higher P-value related to a weak susceptibility? Normally this is found when mean susceptibility is close to "zero", due to a diamagnetic component of the susceptibility balancing out a para-/ferromagnetic component. In this case is it due to the fact that the small sizes leads to a susceptibility that gets within the accuracy range of the bridge?

4) It appears that the grain size of samples is much smaller that the sample size, but is this really the case, i.e., are there enough grains to reflect the anisotropy of a larger volume? I once tried with a shale/slate sample and a biotite crystal to reduce sample size of a cube to see if this affected the AMS. Although I did not go below a 1-cm edge, I did not see an effect. But in these cases, I had very fine grain sizes with respect to

SED
volume or a single crystal.

5) Going back to the point with directional data, how variable is the mineralogy between the different types of samples? How variable is the high-field susceptibility extracted from the magnetization versus field measurements?

In regards to the first point, please see answer above regarding the difference in mineral composition (homogeneity of the mineralogy). The high-field susceptibility varies about an order of magnitude, and is typically higher in the altered pst, which coincides with the low field susceptibility measurements. The Ms is higher in the pristine pseudotachylytes, but this is likely due to the contribution of magnetite in these samples. The high-field susceptibility is presented in Table 2 of the manuscript.
6) Shape is often never a good parameter in looking at deformation, and it is not surprising that the shapes are so variable.

Minor comments 1) The authors mention frequency-dependent susceptibility, but do not mention it further in the text. Did they try to measure the AMS at different frequencies, e.g., in the PST APST samples?
measurements were to identify/investigate the possible contribution of authigenically formed superparamagnetic magnetite in the pseudotachylyte. However, this could not be done based on the results. We have added text to the results section and a new figure, and incorporated these results into the section on results of AMS (the new section is called Anistropy of magnetic susceptibility and frequency dependent susceptibility).

2) I am not sure how significant an isolation of a ferromagnetic component is in the host rock or APST. I would put little faith in trying to extract a saturation magnetization. They are surely artefacts and I would not even show. In this case, it would have been better to measure the acquisition of IRM. One could have probably gotten a convincingly significant signal that would allow for comparison.

Indeed the Ms and Mrs in the host rock and APST are artifacts and we have made a note of this in the in the discussion part on the source of magnetic susceptibility and magnetic fabric (although we kept the images in the figure). We agree that IRM acquisition curves would have provided useful results for comparison and determination of saturation remanent magnetization (as well as coercivity of remanence). Unfortunately, we did not have the possibility to carry out such measurements for the revised manuscript.

Line 18: remove hyphen in information

------ done ------

Line 35: 2.1 instead of 1.1

------ done -------

Lines108-109: in the case that T = 1 or -1, then the ellipsoid is rotationally oblate, respectively prolate. The ellipsoid is still oblate for T> 0, and prolate for T

ellipsoid is oblate for T>0 and prolate for T

Figure 9. Mass dependent susceptibility ( $\chi$ ) measured as a function of frequency. Error bars represent one sigma standard deviation from repeat measurements of bulk magnetic susceptibility ( $8 \ge n \ge 3$ ). Note that the presentation format of data for the different rock types differ compared Figures 7 and 8, which present the mean magnetic susceptibility, km = (k1 + k2 + k3)/3.

SED

---

## Author Response (AR2)

Dear editors, Chris Colletini and Federico Rossetti,

We are pleased to see the decision of the journal. We here give our responses to the comments raised by the editor, which are given in **bold** lettering to help distinguish from the comments of the editor.

5

Thanks for your consideration and with sincere regards, Bjarne Almqvist and Hagen Bender, on behalf of the author team

**10 Two comments were raised by the topical editor, which we address here:**

1) The first paragraph of the abstract is describing something that the Authors are not able to achieve with their dataset since they say that "Analysis of structural and magnetic fabric data yield no kinematic information on the direction and sense of seismic slip". Therefore, I suggest to re-adjust this first paragraph in order to be more consistent with main findings of the manuscript. For example, the Authors can add a sentence clearly stating the

15 general problem being addressed by the study, i.e. difficulties in using magnetic anisotropy from small samples to infer fault kinematics.

We have modified the abstract in the new version of manuscript, which is given at the end of this document, with track changes to indicate the modified parts. We have changed the text mainly to emphasize the findings of sample size and accordingly moved some text, towards the beginning of the abstract. We have added a final sentence to the to conclude our study.

2) Some figures have to be improved. Figure 2: add some details to show: a) the 3 cm wide fault vein, that is foliation-parallel and exhibiting a 5-mm-thick band of bluish, altered pseudotachylyte at its top; b) the brittle, steeply W-dipping normal faults crosscutting the ductile fabric & calcite slickenfibres on some of the fault planes.

25 steeply W-dipping normal faults crosscutting the ductile fabric & calcite slickenfibres on some of the fault planes. In a new figure 2 we have indicated a) the fault vein thickness and b) the steeply dipping normal faults.

Some labelling or numbers in figures 7, 8, 10, 12, 13 are too small.

We have modified the labelling in figures to larger font size and new figures are submitted.

[revised manuscript text omitted]